# Metabolite signatures of diverse *Camellia sinensis* tea populations

Xiaomin Yu [1,10], Jiajing Xiao[2,3,10], Si Chen[1,10], Yuan Yu [1], Jianqiang Ma[4], Yuzhen Lin[1], Ruizi Li[1], Jun Lin[1], Zhijun Fu[1], Qiongqiong Zhou[5], Qianlin Chao[6], Liang Chen [4✉], Zhenbiao Yang [7,8✉] & Renyi Liu [1,9✉]

The tea plant (*Camellia sinensis*) presents an excellent system to study evolution and diversification of the numerous classes, types and variable contents of specialized metabolites. Here, we investigate the relationship among *C. sinensis* phylogenetic groups and specialized metabolites using transcriptomic and metabolomic data on the fresh leaves collected from 136 representative tea accessions in China. We obtain 925,854 high-quality single-nucleotide polymorphisms (SNPs) enabling the refined grouping of the sampled tea accessions into five major clades. Untargeted metabolomic analyses detect 129 and 199 annotated metabolites that are differentially accumulated in different tea groups in positive and negative ionization modes, respectively. Each phylogenetic group contains signature metabolites. In particular, CSA tea accessions are featured with high accumulation of diverse classes of flavonoid compounds, such as flavanols, flavonol mono-/di-glycosides, proanthocyanidin dimers, and phenolic acids. Our results provide insights into the genetic and metabolite diversity and are useful for accelerated tea plant breeding.

[1] FAFU-UCR Joint Center for Horticultural Biology and Metabolomics, Haixia Institute of Science and Technology, Fujian Agriculture and Forestry University, 350002 Fuzhou, China. [2] Shanghai Center for Plant Stress Biology, Chinese Academy of Sciences, 3888 Chenhua Road, 201602 Shanghai, China. [3] University of Chinese Academy of Sciences, 100049 Beijing, China. [4] Key Laboratory of Tea Biology and Resources Utilization, Ministry of Agriculture and Rural Affairs, Tea Research Institute, Chinese Academy of Agricultural Sciences, 310008 Hangzhou, China. [5] College of Horticulture, Henan Agricultural University, 450000 Zhengzhou, China. [6] Wuyi Star Tea Industry Co., Ltd, 354300 Wuyishan, China. [7] Institute of Integrative Genome Biology, University of California at Riverside, Riverside, CA 92521, USA. [8] Department of Botany and Plant Sciences, University of California at Riverside, Riverside, CA 92521, USA. [9] Center for Agroforestry Mega Data Science, Haixia Institute of Science and Technology, Fujian Agriculture and Forestry University, 350002 Fuzhou, China. [10] These authors contributed equally: Xiaomin Yu, Jiajing Xiao, Si Chen. ✉email: liangchen@tricaas.com; yang@ucr.edu; ryliu@fafu.edu.cn

Plants are a rich source for specialized metabolites that are not essential for their growth, development, and reproduction. Based on chemical structures, they are grouped into three major classes: terpenes, phenolics, and nitrogen-containing compounds. It is estimated that 100,000 to one million specialized metabolites are collectively produced by plants and any single plant produces a subset ranging from 5000 to tens of thousands of these metabolites[1–3]. Recent research also suggests that there exists a high level of qualitative and quantitative variations of metabolism within a plant species[4]. Specialized metabolites not only have key roles in plant adaptation to the environment and resistance to biotic and abiotic stresses but also provide natural products used for treating human diseases and important for human health and food quality[3]. More than two-thirds of small-molecule drugs introduced in the last two decades are either plant extracts or their close derivatives[2]. Due to their importance, intensive efforts have been devoted to the dissection of biosynthesis and genetic regulation of plant-specialized metabolites through reverse and forward genetic approaches[4]. In particular, the recent application of genomic, transcriptomic, and metabolomic profiling data makes it possible to not only explore the metabolite diversity between different species and different accessions of the same species, and understand the underlying evolutionary mechanisms[5–7], but also identify candidate regulators through association analyses[8,9]. However, most of the identified candidate metabolic quantitative loci (mQTLs) and regulators lack experimental validation[4] and scientists cannot yet answer important questions such as: what are the underlying evolutionary mechanisms for metabolomic diversity between different species and different accessions of the same species? What metabolites are responsible for flavors of plant food products? How are plant metabolites regulated at the transcriptional, translational, and epigenetic levels? What are the functional roles of structurally similar but distinct metabolites?.

The tea plant [Camellia sinensis (L.) O. Kuntze] is an excellent model system to address these questions due to its high contents and diversity in all three classes of specialized metabolites[10–14]. Tea is the most popular non-alcoholic beverage and offers a plethora of health benefits such as anti-oxidant, anti-cancer, anti-cardiovascular disease and anti-allergic activities[15]. Tea popularity is also attributed to a variety of rich flavors that come from all three classes of specialized metabolites[16]. Among the structurally diverse phytochemicals produced in tea plants, flavonoids such as catechins are best characterized molecularly and biochemically[17–19]. Synthesized through the phenylpropanoid and flavonoid pathways, catechins in tea are a mixture of different enantiomers and their gallic acid conjugates. They are most abundantly detected in tea leaves, among which (−)-epigallocatechin-3-gallate (EGCG) is predominant and the most bioactive[20]. Furthermore, tea plants synthesize a myriad of aroma compounds (e.g., volatile terpenes, fatty acid derivatives, and phenylpropanoids/benzenoids) in response to biotic and abiotic stresses[21]. Last but not the least, caffeine[22] and non-proteinaceous amino acid L-theanine[23,24], which is particularly abundant in tea plants, also are key contributors to tea quality. An important goal for tea improvement is to breed for the increases of specific target metabolites and/or downregulation of some other target metabolites[25]. Comprehensive evaluation of metabolite contents of representative accessions will not only help us identify metabolite properties and signatures of different accessions, but also help us make wise selections of parental lines for tea breeding. Previous studies have revealed the metabolite content differences among different types of tea, but they mostly focused on processed tea products[26–28]. Because the metabolite types and contents change dramatically during tea processing[29–31], the metabolite differences among processed products may not correlate to the genetic

backgrounds of corresponding tea accessions. To date only a very limited number of targeted or untargeted metabolite profilings have been performed on a small number of tea accessions to compare the metabolite contents of fresh tea samples[20,32]. No untargeted metabolomic studies have been carried out on fresh tea leaves from diverse tea populations.

China is likely the center of origin for tea plant, and is the top country for tea cultivation and production, accounting for ~40% of total world production in 2017 (www.fao.org/faostat). China is home to more than 3000 tea accessions, and the genetic and metabolite diversity of tea population in this country largely represents the tea diversity in the world[33]. Modern tea cultivars are derived from hybrids within or between two major tea varieties, the large-leaved C. sinensis var. assamica (CSA) and the small-leaved C. sinensis var. sinensis (CSS). Molecular markers have been used to illustrate the genetic relationship among cultivated tea accessions. For example, Yao et al.[34] used 96 EST-SSR markers to analyze 450 tea accessions in different tea-producing regions in China and found that the cultivated tea accessions could be classified into five groups, clustered roughly around their growing locations. However, a recent study, using 6,252,201 single-nucleotide polymorphism (SNP) markers obtained from genome-resequencing data, separated 81 collected accessions into three clusters (CSS, CSA, and wild type)[35]. The smaller number of clusters revealed by this study is most likely because only 81 accessions were evaluated, among which only 58 were cultivated accessions. To resolve this discrepancy, a comprehensive evaluation of genetic diversity and population structure of a larger number of representative tea accessions, especially cultivated tea accessions, using genome-wide markers such as SNPs is needed.

Recently, the draft genomes of CSA and CSS have been published[18,19,35,36], making it feasible to conduct genome-wide large-scale omics analyses. The tea genome is large (~3.1 GB) and complex, containing at least 34,000 protein-coding genes. Similar to other large plant genomes, the majority (>64%) of the tea genome contains various repetitive elements. The genome sequences not only provide a list of genes that are involved in the biosynthesis of three key compounds, catechins, caffeine, and theanine, and evidence for lineage-specific expansions of genes associated with flavonoid biosynthesis, but also help reveal the variations of metabolites and gene expression among different tissues and among different Camellia species. Comparison of the CSA and CSS genomes indicated that they diverged ~0.38–1.54 million years ago and analysis of genic collinearity showed that the tea genome resulted from two rounds of whole-genome duplications[19]. However, the genetic and metabolite diversity among different tea accessions remains to be explored.

Here we integrate transcriptomic and metabolomic analyses to study the population structure and phylogenetic relationships among major tea cultivars and association of tea metabolites with populations. We chose to use transcriptomic data rather than genome-resequencing data for this work because transcriptomic data can provide sufficient amount of polymorphism markers that are mostly within or around gene-encoding regions, without wasting sequencing power on intergenic regions and with additional benefit of examining gene expression changes. Deep RNA-sequencing is emerging as an important tool for rapid analysis of phylogenetic relationships among cultivars and evolutionary history of the plant kingdom[37–39]. Our comprehensive analyses showed that these representative cultivated tea accessions could be classified into five major groups and each group had unique gene expression and metabolite signatures. Our results provide molecular and metabolic markers for tea breeding, insights into the relationship between tea populations and specialized metabolites, and a foundation for the elucidation of mechanisms

underlying the diversity, high contents, and dynamics of specialized metabolites in tea plants.

## Results

**Phylogenetic relationships among representative Chinese tea accessions.** Over thousands of years, diverse tea cultivars have been derived for various tea flavors and adaption to the environment via a combination of breeding and natural variation, and are mostly propagated through cuttings. However, their origins and phylogenetic relationship remained poorly understood. Here we used RNA-sequencing (RNA-seq) to study the phylogenetic relationships among 136 accessions (128 cultivars) collected from different growing regions, covering all major tea-producing provinces/regions in China (Fig. 1a), such as Yunnan, Fujian, Hunan, Anhui and Zhejiang. Accessions from Yunnan also included a close tea relative (*C. taliensis*). The second leaf samples (three biological replicates from each accession) were collected and subjected to RNA-seq analysis. On average, more than 5 GB of RNA-seq data were generated in each sample after adapter sequence and low-quality bases were removed. After aligning clean reads to the reference genome[19], a total of 925,854 high-quality SNPs were identified, including 320,946 SNPs that are located in protein-coding regions.

We analyzed the phylogenetic relationship and evolutionary history of the 136 collected accessions by using a maximum likelihood-based phylogenetic tree constructed from 45,162 SNPs that are located on fourfold-degenerate sites, using the tea relative accession S159 and four other close relatives of tea plants (RNA-seq data for these tea relatives were collected from published data) as the outgroup (Fig. 1b). According to this phylogenetic tree, the cultivated accession "Chaoyang" (S9) is most closely related to the

tea relatives, consistent with the fact that its parental line, "Chongqing Pipacha", was derived from a wild tea plant from Sichuan Province. The other 134 accessions could then be classified into five major groups, with group 1 containing exclusively CSA accessions or hybrid accessions with a dominant CSA genetic background, such as "Yunkang 10" (S55) and "Yinhong 1" (S116). CSA cultivars are mostly distributed in Yunnan province and are used to mainly process black tea and dark tea such as Pu'er tea. The other four groups contain middle/small-leaved accessions (Fig. 1b).

All accessions included in group 2 were adapted from wild tea plants and propagated asexually. Most of them originated from Hunan, Guangdong, and Chongqing, which may be the natural hybrid zones between CSA and CSS lineages. Group 3 contained mainly hybrid accessions that were generated by breeders or through natural hybridization. "Fuding Dabaicha" and "Fuding Dahaocha", two premium cultivars used to process white and green tea, were included in this group. Moreover, 15 out of 32 accessions in this group descended from "Fuding Dabaicha". For example, "Zhenong 113" (S96) is the hybrid accession derived from the crossing between "Yunnan Dayecha" and "Fuding Dabaicha". Most tea accessions falling into group 4 initiated from geographically close regions in China, such as Zhejiang, Anhui, Jiangxi and Jiangsu, and are most suitable for manufacturing high-quality green tea. Representative examples were "Longjing 43" (S118), "Xicha 5" (S80), "Shifocui" (S103), and "Suchazao" (S78 and S58). Thirty out of 36 tea accessions in group 5 were mainly grown in Fujian and are typically used to process oolong tea. For example, Anxi Tieguanyin, the most well-known oolong tea from Southern Fujian, is processed using the "Tieguanyin" (S44) tea cultivar. Some hybrid accessions such as

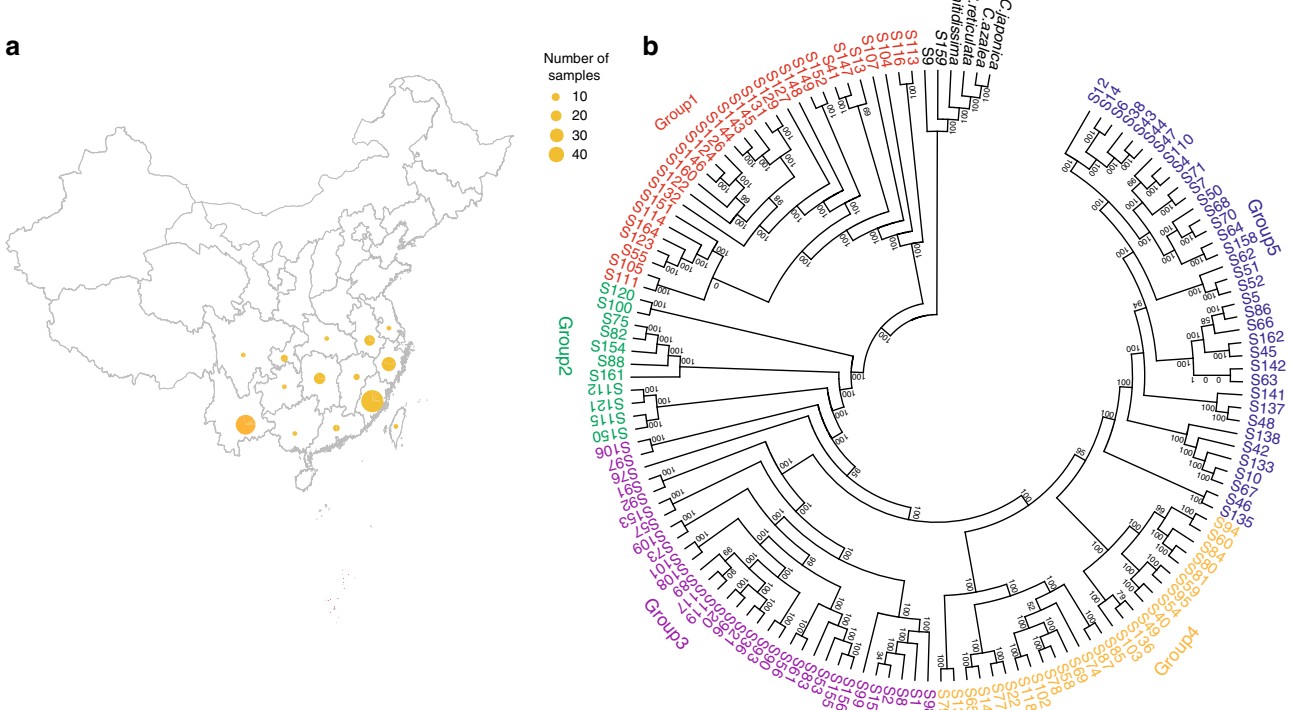

**Fig. 1 Geographic origins and phylogenetic relationships of 136 representative tea plant accessions in China. a** Geographic origins of the tea plant accessions examined in this study. The map of China was generated using the R package "chinamap" (https://github.com/GuangchuangYu/chinamap). **b** An approximate Maximum Likelihood-based phylogenetic tree constructed using 45,162 fourfold-degenerate SNPs that were identified from mapped RNA-sequencing data. The tree was rooted using five tea relative species (in black) as outgroup. Numbers at the branch points represent support values (percentage) based on 1000 bootstrapping replicates. Five main clades were identified and indicated in different colors: group 1 (red), group 2 (green), group 3 (purple), group 4 (yellow), group 5 (blue). Source data underlying **b** are provided as a Source Data file.

| | Group 1 | Group 2 | Group 3 | Group 4 | Group 5 |
|---|---|---|---|---|---|
| **Table 1 Number of signature SNP sites on which major alleles were different between a pair of tea groups.** | | | | | |
| Group 1 | | | | | |
| Group 2 | 6838 | | | | |
| Group 3 | 11,362 | 2617 | | | |
| Group 4 | 13,534 | 2817 | 132 | | |
| Group 5 | 11,762 | 3791 | 674 | 132 | |

"Chuntaoxiang" (S68), "Huangguanyin" (S4), "Huangmeigui" (S110), "Mingke 1" (S6), and "Jinmudan" (S14) were derived from a cross between "Tieguanyin" (S44) and "Huangdan" (S47) and hence were clustered with their parental lines in the phylogenetic tree. Also included in this group were many cultivars like "Rougui" (S133), "Shuijingui" (S137), "Bantianyao" (S138), and "Baijiguan" (S135), which are traditionally used for producing Wuyi Rock tea, a well-known oolong tea from Northern Fujian.

Next, we examined each of the 925,854 high-quality SNP sites and identified signature SNPs that could be used to separate different groups of tea accessions. As shown in Table 1, the largest difference was observed between group 1 and the other groups, differing in 6838–13,534 SNP locations. In contrast, the numbers of SNP sites that separated the remaining groups were much smaller, reflecting a closer relationship among middle/small-leaved tea accessions. Notably, on 8187 SNP sites, group 1 possessed different major alleles from the other four groups, representing genetic divergence between CSA and CSS lineages. Signature SNPs were also found on genes that are known to be involved in the biosynthesis of characteristic metabolites in tea. For example, two genes involved in catechin and caffeine biosynthesis, namely, *LAR* (TEA027582, encoding a leucoantho-cyanidin reductase) and *TCS* (TEA015791, encoding a caffeine synthase), contained several SNP sites on which the major alleles varied among different groups, with some being non-synonymous (Supplementary Fig. 1). Non-synonymous muta-tions cause changes in protein sequence, and could change protein function/enzyme activity. LAR is an important enzyme in the catechin biosynthesis pathway and is responsible for converting leucocyanidin/leucodelphindin to C/GC. Previous evolutionary analysis showed that extensive sequence diversity exists among *LAR* genes in different plants as well as among three *LAR* homologs in tea[40]. TEA026582 exhibited more than 10-fold higher expression than the other two *LAR* homologs in our samples and thus likely had a major functional role. Although the two non-synonymous SNPs that we discovered here are not located within the three well-conserved LAR-specific motifs (RFLP, ICCN, and THD) or at the known substrate-binding sites[40], they may still cause change in enzyme activity and thus are worth further investigation. Tea caffeine synthase (TCS) is a *N*-methyltransferase that converts 7-methylxanthine to theobro-mine (theobromine synthase or TS activity) and theobromine to caffeine (caffeine synthase or CS activity) and has a major role in the caffeine synthesis pathway[41]. There are 11 TCS homologs in the tea genome, among which TEA015791 (*TCS1*) has the highest expression level and has a predominant role. It has been shown that there are at least six natural *TCS1* alleles (named *TCS1a* to *TCS1f*, respectively) in different tea accessions[41]. Furthermore, different *TCS1* alleles exhibit significant difference in TS and CS activities. While *TCS1a* and *TCS1d-f* have both TS and CS activities, *TCS1a* has a lower CS/TS ratio than *TCS1d-f*[41]. *TCS1b-c* have only TS activity, resulting high level of theobromine and almost no caffeine in the two "caffeine-free" wild tea accessions "Hongyacha" and Cocoa tea (*C. ptilophylla* Chang)[42]. Amino acid residue variations on our non-synonymous SNP sites indicate that in CSS accessions *TCS1a* is the dominant *TCS1* allele, whereas in CSA accessions, a *TCS1d/f*- like allele is the major allele. The enzyme activity difference of the major *TCS1* allele may affect the theobromine and caffeine accumulation levels in different tea accessions.

**Population structure of tea accessions in China**. To further understand the genetic diversity of tea accessions in China, a Bayesian inference of population structure was conducted using the STRUCTURE software[43]. The optimal number of clusters ($K$) was determined to be 5 by Harvester[44] (Fig. 2a), indicating that the sampled tea accessions in China could be grouped into five populations, in agreement with the phylogenetic analysis. With $K = 2$, 134 collected accessions were classified into a large-leaved tea population and a middle/small-leaved tea population, indicating the apparent genetic divergence between CSA and CSS. With the optimal $K$ being 5, population 1 mainly consisted of accessions in group 1 and several other accessions that would be classified into other groups mainly due to controlled crossing and/or selection from natural hybridization. Population 2 mainly contained accessions from group 2 and several other accessions that came from natural hybridization between CSA and CSS. Population 3 only included accessions in group 5, most of which were derived from "Tieguanyin" and "Huangdan". Population 4 mainly con-tained accessions clustered in group 4 and several oolong tea accessions. Population 5 mainly contained accessions in group 3 derived from the hybridization breeding process between CSA and CSS. Therefore, results from population structure analysis agreed well with those from the phylogenetic analysis: the representative tea accessions in China could be classified into five groups/populations. Principal component analysis (PCA) of the genetic distance of these accessions using the SNP data also illustrated that accessions in each of the five groups/populations were naturally clustered together (Fig. 2b).

**Identification of regions that were potentially subject to selective sweeps**. To illustrate the genetic differentiation among five tea groups, we calculated the average $F$st values in pairwise comparisons of tea groups (Fig. 3a). The genetic differentiation between CSA and any of the other four groups were significantly ($p < 0.001$) higher than those between any pair of subpopulations within CSS, supporting the apparent genetic divergence between CSA and CSS. The lowest genetic differentiation was detected between group 4 and group 5, indicating the relatively close genetic relationship between green tea and oolong tea accessions.

During the evolution and domestication of tea plants, some genomic regions may have been subject to selective sweep because such regions contain genes that were related to traits selected by natural environments or by artificial breeding. The XP-CLR software[45] was used to identify potential selective sweep regions by comparing non-overlapping 10 kb regions along the tea genome between any two of the aforementioned five groups of tea accessions that we identified through phylogenetic analysis

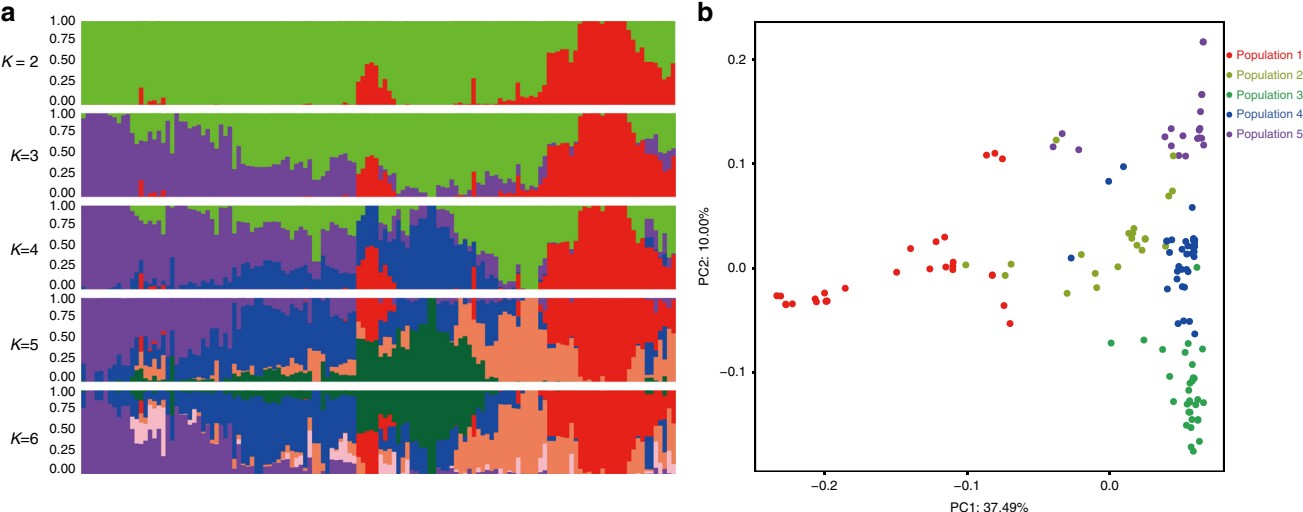

**Fig. 2 Population structure of 134 representative tea plant accessions in China. a** Model-based clustering analysis with different $K$-values (number of clusters). The optimal $K$-value is 5 as determined by the Harvester software. **b** PCA analysis using SNP allele data showing the genetic distances among tea plant accessions in five groups identified in **a**. Source data are provided as a Source Data file.

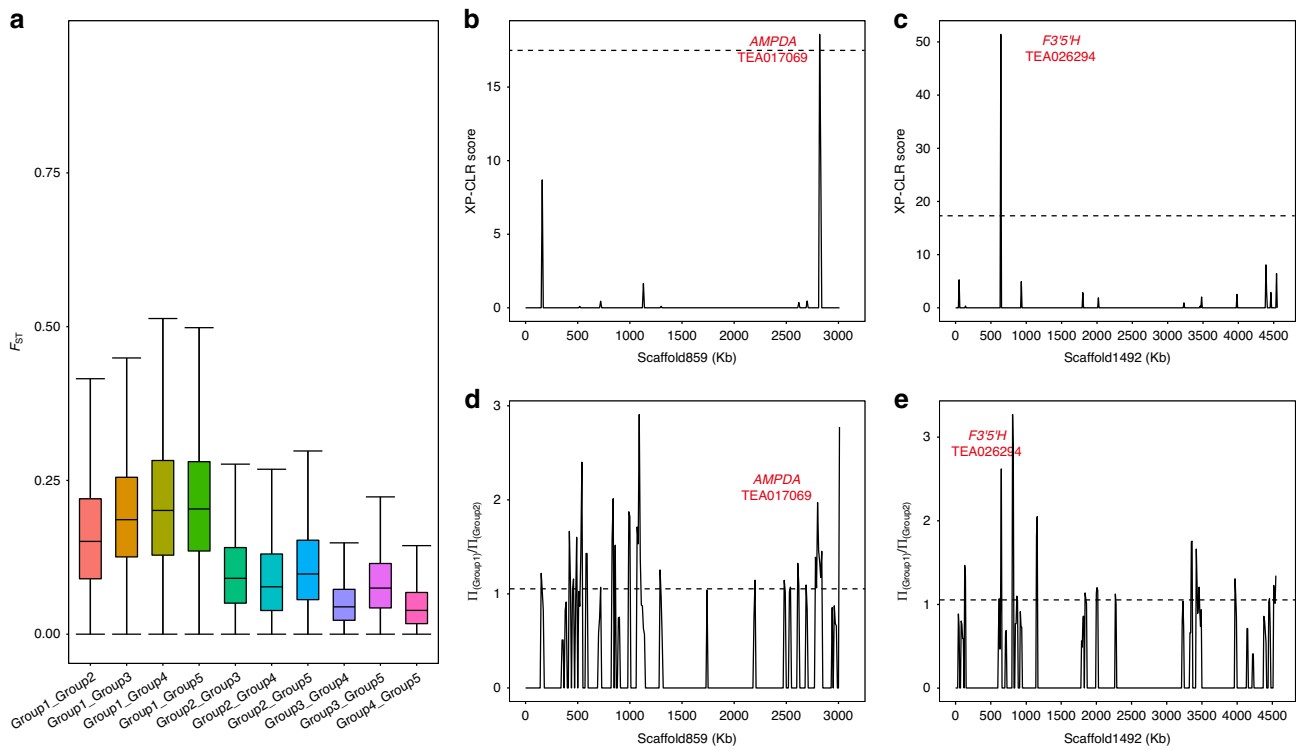

**Fig. 3 Population differentiation and selective sweep regions across five groups of tea plant accessions. a** Boxplot of Fst values calculated from pairwise comparisons of five groups of tea plant accessions (identified in Fig. 1a). Fst was calculated on 100 kb sliding windows with a step size of 10 kb along all scaffolds in the tea reference genome. ($n = 180,916, 181,969, 182,127, 182,256, 181,842, 181,136, 181,436, 181,989, 182,235, 181,300$ sliding windows for plots ordered from left to right). Boxes = interquartile ranges, middles = medians, whiskers = 1.5 × the interquartile range. **b–e** Plot of selective sweep (**b**, **c**) and nucleotide diversity values (**d**, **e**) along scaffold859 and scaffold1492 that contain selective sweep regions. The dotted line in **b**, **c** indicates a threshold value of 18.40 and the dotted line in **d**, **e** indicates a threshold value of 1.054. Source data are provided as a Source Data file.

(Fig. 1b). We identified 833 potential selective sweep regions that contained 1132 genes.

Two examples of selective sweep regions were shown in Fig. 3b–e. A selective sweep region on scaffold859 contained the *AMPDA* gene (TEA017069, encoding an adenosine monophosphate deaminase) that is involved in caffeine biosynthesis. As one of the early steps in caffeine biosynthesis, AMPDA converts

adenosine monophosphate (AMP) to inosine monophosphate (IMP), freeing an ammonia molecule in the process. Another gene, *F3′5′H* (TEA026294, encoding a flavonoid 3′,5′-hydroxylase), is involved in the biosynthesis of catechin, and was recently identified to have a role in governing the ratio of di/tri-hydroxylated catechins and catechin contents[46]. These results suggested that some metabolic pathways may have been subject to

**Table 2 Total number of detected, differentially accumulated, and signature metabolites identified in this study.**

| Mode | Total[a] | DAMs[b] | Signature metabolites in group 1[c] | Signature metabolites in group 2[c] | Signature metabolites in group 3[c] | Signature metabolites in group 4[c] | Signature metabolites in group 5[c] |
|------|----------|---------|-------------------------------------|-------------------------------------|-------------------------------------|-------------------------------------|-------------------------------------|
| POS  | 2672     | 129     | 15                                  | 3                                   | 0                                   | 0                                   | 1                                   |
| NEG  | 1997     | 199     | 21                                  | 9                                   | 0                                   | 1                                   | 4                                   |

[a]Total number of metabolic features detected
[b]Number of differentially accumulated metabolites (DAMs).
[c]Number of metabolites that showed significantly higher accumulation in one group of tea accessions than any of the other four tea groups.

strong selection during the evolution and domestication of tea and some specialized metabolites may have been key traits for breeding and domestication.

**Identification of tea metabolites via untargeted metabolomic analyses of different tea populations**. As a first step in assessing how tea metabolites are related to diverse genetic backgrounds, we sought to expand our knowledge of tea metabolites by untargeted metabolomic analysis. Previous untargeted analyses were limited to processed tea products or fresh leaves of a small group of closely related tea cultivars[20,26,28,30,47,48]. We anticipate that many tea metabolites were missed in these analyses. Hence we analyzed the metabolite contents of fresh tea leaves in the 136 representative tea accessions using untargeted metabolomics analysis of the second leaf samples by ultra-performance liquid chromatography-quadrupole time-of-flight mass spectrometry (UPLC-QTOF MS). In total, 2672 and 1997 mass/retention time features were detected in positive (POS) and negative (NEG) electrospray ionization (ESI), respectively. To remove ultra-low abundance signals, ions with a normalized relative abundance lower than 500 in all accessions were filtered, leaving 752 and 503 metabolic features in respective modes for further analysis (Table 2 and Supplementary Data 1, 2). Putative metabolite identity was assigned for tea leaf constituents in accordance with databases and literatures, and with comparison with authentic standards (Supplementary Data 3 and 4).

Like other plant species, tea plants possess their own specialized metabolome comprised of many isomeric compounds (Supplementary Data 3 and 4). Given their similar/same MS/MS fragmentation behaviors and small differences in retention time, resolving these structural isomers by the MS-based metabolomics approach alone still remains challenging and may necessitate the use of nuclear magnetic resonance (NMR) for further unambiguous structural elucidation. Nevertheless, we found that flavonol glycosides (in the RT window of 5.1–11.3 min and mostly between 7 and 11 min) and proanthocyanidins (in the RT window of 2.4–10.1 min and mostly between 4 and 9 min) were the two most structurally diverse classes of specialized metabolites detected in the methanol extracts of fresh tea leaves. For example, three compounds (RT = 9.00, 9.33, and 9.59 min) showing [M−H]$^-$ at $m/z$ 635.1609, 635.1615, and 635.1608, respectively, were all putatively identified as kaempferol acetylhexose deoxyhexose. Two doubly charged ions (RT = 10.41 and 11.07 min) with their corresponding singly charged ions around $m/z$ 1031.30 were found to be another pair of isomers. They were putatively characterized as isomers of kaempferol 3-(p-coumaroyl-rhamnosyl)rutinoside-7-rhamnoside, whose occurrence in tea plants has not been documented. Six compounds (RT = 4.11, 4.35, 4.44, 4.80, 4.93, and 5.78 min) that were predicted to have the same formula $C_{30}H_{26}O_{13}$ shared similar fragment ions and were putatively assigned as isomers of EC-GC dimer. Likewise, in ESI$^-$, two metabolites eluted at 5.23 and 5.75 min, respectively, were shown to have the same formula $C_{52}H_{42}O_{25}$. Given the same MS/MS fragmentation ions generated, we believe that again they

were isomers, and based on the match in the Dictionary of Natural Products database (http://dnp.chemnetbase.com), we tentatively assigned them as two galloylated trimeric proanthocyanidins. Metabolites with the same formula have not been previously reported in tea plants. The RT window between 11 and 17 min was occupied primarily by triterpenoid saponins and terpenoid glycosides (albeit in low abundance), many of which have rarely been described as constituents in fresh tea leaves. Another metabolite that immediately catches our attention eluted at 5.26 min and had $m/z$ at 225.0972 in ESI$^+$. The deduced formula was $C_9H_{12}N_4O_3$, matching that of theacrine, a caffeine-like xanthine alkaloid that has so far only been reported in C. assamica var. Kucha[49]. The biosynthesis of theacrine has sparked a lot of research interest since theacrine is non-stimulatory and hence may guide the development of decaffeinated drinks[49]. To our great interest, out of all the tea resources that we screened, only "Nannuoshan Dayecha 3" (S131) was found to produce a large amount of theacrine. This tea accession adds another genetic resource, besides Kucha, to facilitate mechanistic investigations into how caffeine is transformed into theacrine.

**Metabolite signatures for the five different tea phylogenetic groups**. Global clustering analysis of the tea metabolite profiles described above suggests that genetic background had a stronger effect on metabolite contents than environment factors because tea accessions in the same phylogenetic group were more likely to cluster together than those from the same growing location (Supplementary Fig. 2a, b). As described above, different tea plant phylogenetic groups/populations more or less represent their processing suitability (e.g., group 4 accessions are commonly used for making green tea, whereas group 5 accessions are typically used for making oolong tea). Hence, we were interested in understanding the metabolic basis of the groupings. Specialized metabolites are the primary factors in determining health benefits, tastes and aroma of tea products. To identify metabolite signatures that would ultimately assist in breeding, we performed pairwise comparisons to detect metabolic features that were differentially accumulated among five groups of tea accessions. 409 and 325 metabolic features were found to be differentially accumulated in at least one pairwise comparison under POS and NEG modes, respectively. Not surprisingly, the highest number was found in group 1 under both modes (Fig. 4a, b), indicating that CSA tea accessions had a quite different metabolite profile from that of CSS tea accessions, consistent with the result from clustering analysis. After careful inspection of raw spectral data, 280 (68%) and 126 (39%) features under POS and NEG modes, respectively, were found to be fragment ions and removed from further analysis. The remaining 129 and 199 features were classified as differentially accumulated metabolites (DAMs), with 75 DAMs being detected in both modes. Among these DAMs, 108 (84%) and 171 (86%) in the respective mode could be confidently (with reference to authentic standards) or putatively assigned to known metabolites (Supplementary Data 3–6).

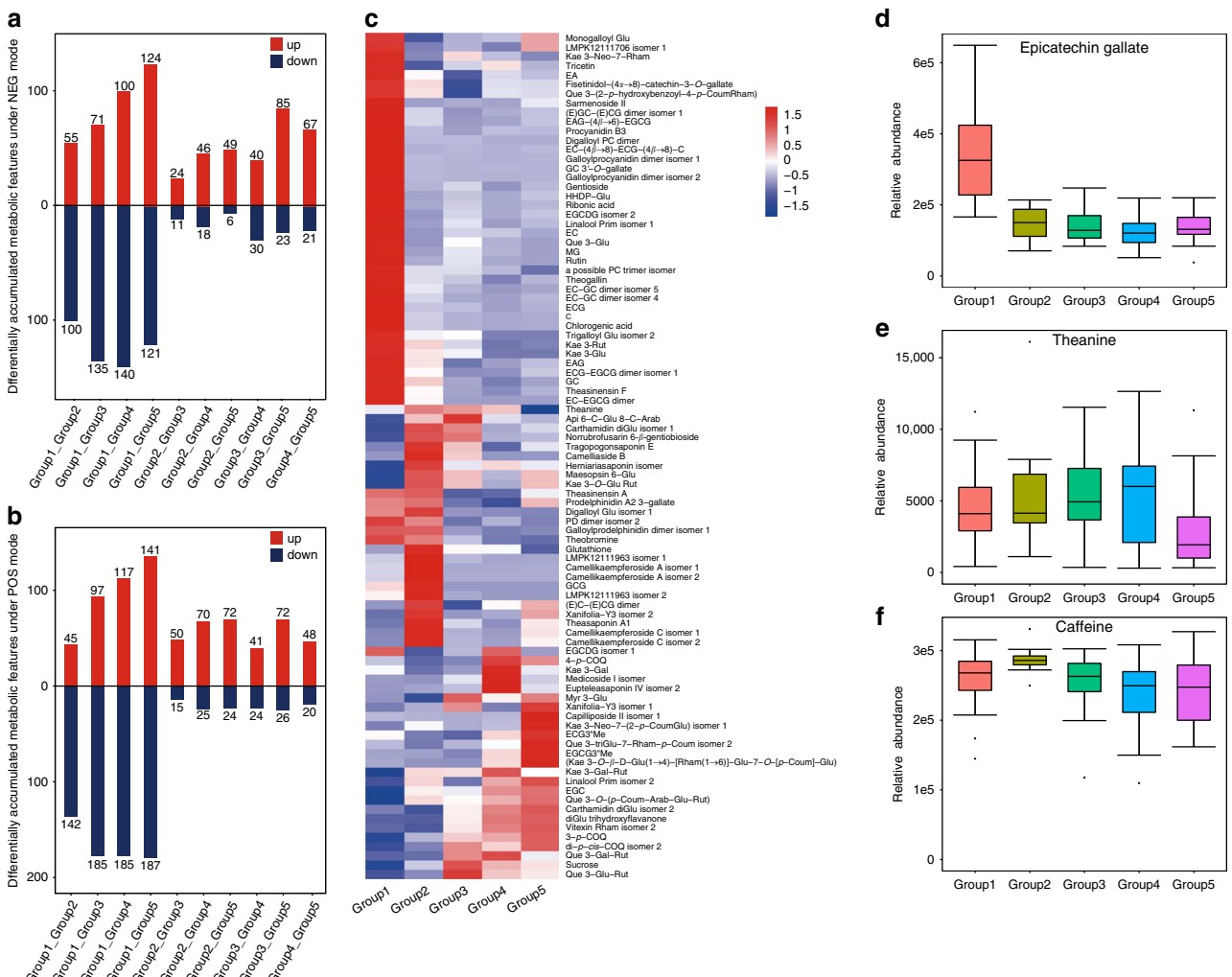

**Fig. 4 Metabolites that showed significant changes in concentration in pairwise comparisons of five groups of tea accessions. a, b** Number of metabolic features that were detected under NEG (**a**) and POS (**b**) modes, respectively, and were identified as differentially accumulated in pairwise comparisons of five groups of tea accessions. Red and blue bars indicate the numbers of metabolic features showing increase and decrease in concentrations, respectively. **c** Heatmap showing the abundance patterns of annotated metabolites with an average relative abundance greater than 500 in at least one tea group and with significant changes in abundance in at least one comparison. **d–f** Box plots of abundance of epicatechin gallate, theanine and caffeine in different tea groups. ($n = 29$ for Group 1, 11 for Group 2, 32 for Group 3, 26 for Group 4, and 36 for Group 5). Boxes = interquartile ranges, middles = medians, whiskers = 1.5 × the interquartile range, single points = outliers. Source data underlying **c–e**, and **f** are provided as a Source Data file.

Clustering analysis of the annotated DAMs showed that 40 metabolites, mostly a wide range of flavonoids, were highly accumulated in group 1 but in general were lowly accumulated in other groups. Many flavanols (C, EC, GC, ECG, epigallocatechin digallate, gallocatechin 3′-O-gallate, epiafzelechin 3-gallate, and epiafzelechin) were more abundantly accumulated in group 1 (Fig. 4c and Supplementary Data 7). For instance, almost all accessions in group 1 exhibited a significantly higher content of ECG. The mean abundance of this compound in group 1 was at least 2.2-fold that of the other groups (Fig. 4d). This result partly agrees with a previous targeted analysis of catechin contents in representative Chinese tea germplasms, where the accumulation levels of total catechin, ECG, EC, and C were found to be significantly higher in CSA tea accessions than in CSS tea accessions[50]. In addition, mono-/di-/triglycosides of quercetin and kaempferol, proanthocyanidin dimers, hydrolysable tannins and quinic acid derivatives had higher abundance in group 1. These results suggest that the genes involved in the phenylpropanoid/flavonoid pathways may be upregulated in the CSA lineage. Theobromine, the second most abundant purine alkaloid

in fresh tea leaves, was also enriched in this group, with a similar mean content observed in accessions from group 2 (Fig. 4c).

Higher accumulation of kaempferol glucosylrutinoside, two acylated kaempferol tetraglycosides, four acylated kaempferol triglycosides and five triterpenoid saponins were found in group 2. Myricetin 3-glucoside and quercetin 3-O-glucosylrutinoside were more enriched in group 3. Kaempferol 3-O-galactoside, kaempferol 3-O-galactosyl rutinoside, quercetin 3-O-galactosyl rutinoside and two triterpenoid saponins appeared to be enriched in group 4, although the identities of these two metabolites need to be further confirmed by spectroscopic methods. Finally, methylated catechins (EGCG3″Me and ECG3″Me) and coumaroyl derivatives of quercetin and kaempferol tri-/tetraglycosides were specifically present in higher levels in group 5. However, the level of theanine was apparently lower in group 5 than in other groups (Fig. 4e). Some metabolites may be accumulated at a higher level in more than one groups. For examples, one galloylprodelphinidin dimer, prodelphinidin A2 3-gallate and theasinensin A were present at higher levels in groups 1 and 2. One carthamidin diglucoside, norrubrofusarin 6-β-gentiobioside

as well as apigenin 6-*C*-glucoside 8-*C*-arabinoside occurred more abundantly in groups 2 and 3. Some metabolites such as EGC, diglucopyranosyl trihydroxyflavanone and vitexin rhamnoside were found to have a higher accumulation in tea accessions from groups 4 and 5 (Fig. 4c). On the other hand, the accumulation levels of caffeine did not show significant difference among different groups (Fig. 4f), consistent with the fact that all these tea accessions have been domesticated and/or selected for tea production. In addition, the accumulation levels of EGCG, some proanthocyanidins (one galloylprodelphinidin dimer, procyanidin B2, and one procyanidin trimer), and some other metabolites (e.g., 5-coumaroylquinic acid and linalool oxide primeveroside) appeared to be stable among different tea groups.

Next, we set to detect the signature metabolites whose concentration in one tea group was significantly higher than that in any of the other tea groups. A total of 40 annotated signature metabolites were found. Among them, 36 metabolites were annotated with matches to metabolite databases or authentic standards and the elemental compositions for the remaining four were calculated based on the accurate mass values (Table 3). Group 1 had the highest number of signature metabolites under both POS and NEG modes, whereas group 3 had no signature metabolite, probably because tea accessions in group 3 resulted from hybrid breeding between the two major types of tea (CSA and CSS). Among the annotated signature metabolites detected under the NEG mode (Table 3), five flavonol glycosides (rutin, sarmennoside II, quercetin 3-*O*-glucoside, kaempferol 3-neohesperidoside-7-rhamnoside, and kaempferol 3-(4″-(*E*)-*p*-coumaroylrobinobioside)-7-rhamnoside isomer), five proanthocyanidins (two EC-GC dimers, procyanidin B3, galloylprocyanidin dimer, and digalloylprocyanidin dimer), three phenolic acids (theogallin, methylgallate and chlorogenic acid), four flavanols (C, GC, EC, and ECG), one terpenoid glycoside (linalool primeveroside isomer), one ellagitannin (hexahydroxydiphenoyl-glucose), one hydrolysable tannin (trigalloylglucose) and one sugar acid derivative (likely ribonic acid) were highly enriched in CSA accessions (group 1). In contrast, group 2 consistently exhibited the enrichment of four complex coumaroylated kaempferol glycosides (two camellikaempferoside C isomers and two camellikaempferoside A isomers), two triterpenoid saponins (theasaponin A1 and tragopogonsaponin E), and one hydrolysable tannin (digalloylglucose). The signature metabolite for group 4 was a putative triterpenoid saponin (eupteleasaponin IV isomer). Group 5 had high accumulation of two methylated catechins (EGCG3″Me and ECG3″Me) and one triterpenoid saponin (xanifolia-Y3 isomer). The signature metabolites detected under the POS mode in general agreed with those under the NEG mode.

We also identified three (8.07 min_367.0126 *m/z*, 13.40 min_467.1343 *m/z*, and 16.41 min_1189.5400 *m/z*) and two signature metabolites (13.40 min_457.1377 *m/z* and 16.89 min_731.4146 *m/z*) that were unannotated in NEG and POS modes, respectively. For example, a metabolite eluted at 16.41 min with *m/z* 1189.5400 in NEG mode was much enriched in group 2 tea accessions, with an abundance at least 2.8-fold of that in other groups. Another unannotated metabolite eluted at 8.07 min with *m/z* 367.0126 in NEG mode was only detected in groups 2 and 3 and its average abundance in group 2 was 14-fold higher than that in group 3. In POS mode, the concentration of an unknown metabolite (*m/z* = 731.4146, RT = 16.89 min) in group 2 was at least 2.5-fold of that in other groups of accessions. These unannotated signature metabolites deserve further investigation.

It is well-known that the accumulation of specialized metabolites is highly impacted by environmental factors. Thus, the aforementioned existence of signature metabolites for each group of tea accessions collected from diverse locations and environments is quite significant, as it suggests that these signature metabolites are predominantly determined by genetic factors.

**Differentiation of gene expression profiles among different tea populations**. To assess whether the metabolite signatures described above are associated with the transcription of specific genes, we analyzed the difference in gene expression in the second leaves among different tea accessions. We first performed *t*-Distributed Stochastic Neighbor Embedding (*t*-SNE) analysis of the global gene expression profiles. The results showed that the expression profiles of tea accessions in group 1 tended to cluster together, but not for tea accessions in the other four groups (Supplementary Fig. 3a), suggesting that on the one hand, CSA and CSS lineages have significant divergences in gene expression; on the other hand, genetic background is not the sole deciding factor. In particular, the gene expression profiles of tea samples collected from Jiamu Yeyatang tea plantation, which is located in a mountainous region in Yunnan Province with high elevation (~2000 m) and low temperature (average annual temperature is ~17 °C, and average temperature in the coolest month, January, is ~9 °C), form its own cluster (Supplementary Fig. 3b), regardless of their genetic backgrounds, indicating that environmental factors may greatly influence the gene expression profile in tea plants. Clustering analysis using the expression profiles of genes involved in the biosynthesis of catechins, caffeine and theanine also gave similar results, indicating the biosynthetic genes of these metabolites may show differential expression depending on both genetic backgrounds and growing environments. To further illustrate this point, we compared the overall gene expression profiles of five sets of sample groups that are of the same genotype but were grown in different locations. We found that the four "Yunkang 10" samples (S55, S114, S123, and S164) did not cluster together in the *t*-SNE figures (Supplementary Fig. 3a). Twenty-eight out of 33 genes that showed high correlations with the first dimension in *t*-SNE analysis, significantly changed their expression in at least one pairwise comparison in these four samples. In comparison with the other three tea accessions, three genes, namely TEA016601 (*FLS*), TEA023333 (*CHS*), and TEA023790 (*F3′H*), were significantly downregulated in tea sample S164 collected from Jiamu Yeyatang tea plantation, suggesting the downregulation of these three genes in S164 was mainly caused by environmental factors (high elevation and low temperature).

To further compare the gene expression profiles in five groups of tea plants, we performed pairwise comparisons of gene expression levels in different tea groups and identified differentially expressed genes (DEGs), after removing accessions collected from the Jiamu Yeyatang tea plantation. In total, 7674 DEGs were found in at least one comparison. As shown in Fig. 5a, the number of DEGs was apparently lower in the pairwise comparisons among groups 3, 4, and 5 than when any of them was compared to group 1 or 2, suggesting that the gene expression profiles in groups 3, 4, and 5 were more similar to each other. Among the identified DEGs, 31 genes involved in catechin biosynthesis were found to be differentially expressed in at least one pairwise comparison. Similarly, nine and five DEGs were found in the caffeine and theanine biosynthetic pathways, respectively (Fig. 5b), indicating the expression of structural genes involved in the biosynthesis of these metabolites were dynamically regulated and may, at least in part, be attributed to the genetic backgrounds of tea accessions. Clustering analysis of the expression levels of these genes indicated that the expression levels of seven caffeine biosynthetic genes were in general higher in group 1, while many catechin biosynthetic genes were highly expressed in oolong tea cultivars (group 5) (Fig. 5b). For example,

**Table 3 List of annotated signature metabolites of five tea groups.**

| Signature group | Metabolites detected | Tentative metabolite identification | Metabolite class | Mean abundance in group 1 | Mean abundance in group 2 | Mean abundance in group 3 | Mean abundance in group 4 | Mean abundance in group 5 |
|---|---|---|---|---|---|---|---|---|
| Group 1 | 1.93_481.0621 m/z | HHDP-glucose | Ellagitannins | 667.5 | 18.8 | 107.9 | 28.5 | 51.4 |
| Group 1 | 2.06_167.0549 m/z | Ribonic acid | Sugar acids and derivatives | 1131.7 | 30.8 | 71.5 | 23.2 | 44.7 |
| Group 1 | 2.89_344.0743n | Theogallin | Phenolic acids | 31,146.8 | 14,975.9 | 12,871.7 | 8404.9 | 12,089.9 |
| Group 1 | 3.83_306.0740n | Gallocatechin | Flavanols | 16,166.9 | 7079.3 | 6195.0 | 5533.4 | 6741.1 |
| Group 1 | 4.80_593.1297 m/z | EC-GC dimer isomer 4 | Proanthocyanidins | 1176.5 | 491.2 | 377.7 | 360.7 | 454.4 |
| Group 1 | 4.93_593.1293 m/z | EC-GC dimer isomer 5 | Proanthocyanidins | 1600.1 | 566.2 | 396.9 | 323.0 | 412.8 |
| Group 1 | 5.11_577.1348 m/z | Procyanidin B3 | Proanthocyanidins | 1201.0 | 307.4 | 235.4 | 351.5 | 341.6 |
| Group 1 | 5.34_290.0791n | Catechin | Flavanols | 24,258.8 | 4032.7 | 3625.6 | 3542.0 | 3670.3 |
| Group 1 | 5.36_183.0298 m/z | Methylgallate | Phenolic acids | 4633.4 | 1208.2 | 1888.1 | 1306.2 | 1000.6 |
| Group 1 | 5.51_354.0948n | Chlorogenic acid | Phenolic acids | 6272.5 | 803.9 | 265.5 | 265.9 | 134.6 |
| Group 1 | 5.83_747.1552 m/z[a] | EC-EGCG dimer | Proanthocyanidins | 3602.3 | 907.5 | 512.1 | 517.6 | 588.6 |
| Group 1 | 6.25_290.0794n | Epicatechin | Flavanols | 116,557.4 | 48,408.2 | 57,998.7 | 57,396.7 | 52,306.4 |
| Group 1 | 6.57_551.1402 m/z | Gentioside | Xanthone glycosides | 735.6 | 137.8 | 248.7 | 171.1 | 242.1 |
| Group 1 | 6.63_318.0476n | Trigalloylglucose isomer 2 | Hydrolysable tannins | 9744.1 | 3327.0 | 3954.0 | 1512.2 | 1156.8 |
| Group 1 | 7.69_609.1438 m/z | Rutin | Flavonol glycosides | 11,904.4 | 1726.1 | 2665.6 | 1859.8 | 1404.2 |
| Group 1 | 6.78_729.1455 m/z | Galloylprocyanidin dimer isomer 1 | Proanthocyanidins | 8493.2 | 1435.0 | 1174.9 | 1070.7 | 1255.0 |
| Group 1 | 7.85_442.0901n | Epicatechin 3-O-gallate | Flavanols | 334,280.4 | 151,298.2 | 137,580.1 | 121,634.2 | 137,860.9 |
| Group 1 | 7.88_303.0485 m/z[a] | Tricetin | Flavones | 4889.7 | 818.6 | 1984.6 | 2437.5 | 1774.5 |
| Group 1 | 8.00_464.0948n | Quercetin 3-O-glucoside | Flavonol glycosides | 7216.1 | 1158.4 | 2866.5 | 1918.1 | 1710.0 |
| Group 1 | 8.26_739.2086 m/z | Kaempferol 3-neohesperidoside-7-rhamnoside | Flavonol glycosides | 1035.9 | 1.2 | 356.9 | 20.5 | 0.4 |
| Group 1 | 8.42_594.1530n[a] | Kaempferol 3-O-rutinoside | Flavonol glycosides | 3885.6 | 1526.3 | 874.5 | 431.7 | 497.0 |
| Group 1 | 9.37_441.0818n | Digalloylprocyanidin dimer | Proanthocyanidins | 504.5 | 116.2 | 117.1 | 85.6 | 93.7 |
| Group 1 | 10.24_451.1235n | Sarmenoside II | Flavonol glycosides | 1481.6 | 0.0 | 22.2 | 0.5 | 98.1 |
| Group 1 | 10.55_443.1260n | Kaempferol 3-(4''-(E)-p-coumaroylrobinobioside)-7-rhamnoside isomer 1 | Flavonol glycosides | 612.2 | 5.4 | 6.0 | 0.4 | 93.6 |
| Group 1 | 11.20_448.2300n | Linalool primeveroside isomer 1 | Terpenoid glycosides | 1807.1 | 549.0 | 765.8 | 559.3 | 691.2 |
| Group 2 | 5.01_242.0423n | Digalloylglucose isomer 1 | Hydrolysable tannins | 2155.5 | 9033.6 | 718.2 | 368.6 | 218.5 |
| Group 2 | 10.48_517.1448n | Camellikaempferoside C isomer 1 | Flavonol glycosides | 113.8 | 4182.1 | 449.6 | 327.3 | 1193.9 |
| Group 2 | 10.74_517.1448n | Camellikaempferoside C isomer 2 | Flavonol glycosides | 5.7 | 1459.3 | 116.9 | 90.2 | 358.9 |
| Group 2 | 10.90_436.1180n | Camellikaempferoside A isomer 1 | Flavonol glycosides | 113.9 | 1616.3 | 59.5 | 45.3 | 54.1 |
| Group 2 | 11.13_871.2298 m/z | Camellikaempferoside A isomer 2 | Flavonol glycosides | 23.0 | 518.4 | 7.8 | 5.0 | 8.3 |
| Group 2 | 11.33_1189.5616 m/z | Theasaponin A1 | Triterpenoid saponins | 10.9 | 623.5 | 73.6 | 52.3 | 193.7 |
| Group 2 | 16.41_1189.5400 m/z | $C_{60}H_{86}O_{24}$ | Unknown | 14.9 | 677.3 | 72.4 | 21.0 | 237.3 |
| Group 2 | 16.80_1131.5352 m/z[a] | Tragopogonsaponin E | Triterpenoid saponins | 122.9 | 1685.5 | 375.2 | 34.5 | 350.9 |
| Group 2 | 16.89_731.4146 m/z[a] | $C_{44}H_{58}O_9$ | Unknown | 74.1 | 1173.1 | 431.9 | 347.6 | 461.7 |
| Group 4 | 16.36_1157.5720 m/z | Eupteleasaponin IV isomer 2 | Triterpenoid saponins | 0.0 | 1.6 | 52.9 | 940.7 | 19.7 |
| Group 5 | 7.42_472.1004n | EGCG3''Me | Flavanols | 577.4 | 436.9 | 199.8 | 3086.1 | 8095.4 |
| Group 5 | 8.89_456.1053n | ECG3''Me | Flavanols | 227.9 | 187.4 | 46.2 | 784.1 | 1666.9 |
| Group 5 | 13.40_451.1377 m/z[a] | $C_{25}H_{32}O_8$ | Unknown | 65.0 | 158.5 | 307.6 | 376.0 | 821.8 |
| Group 5 | 13.40_467.1343 m/z | $C_{25}H_{24}O_9$ | Unknown | 7.2 | 144.3 | 360.7 | 464.8 | 1051.2 |
| Group 5 | 16.41_1125.5460 m/z | Xanifolia-Y3 isomer 1 | Triterpenoid saponins | 13.3 | 127.7 | 152.8 | 7.4 | 617.5 |

[a]Detected in ESI+. Other metabolites were detected in ESI−.

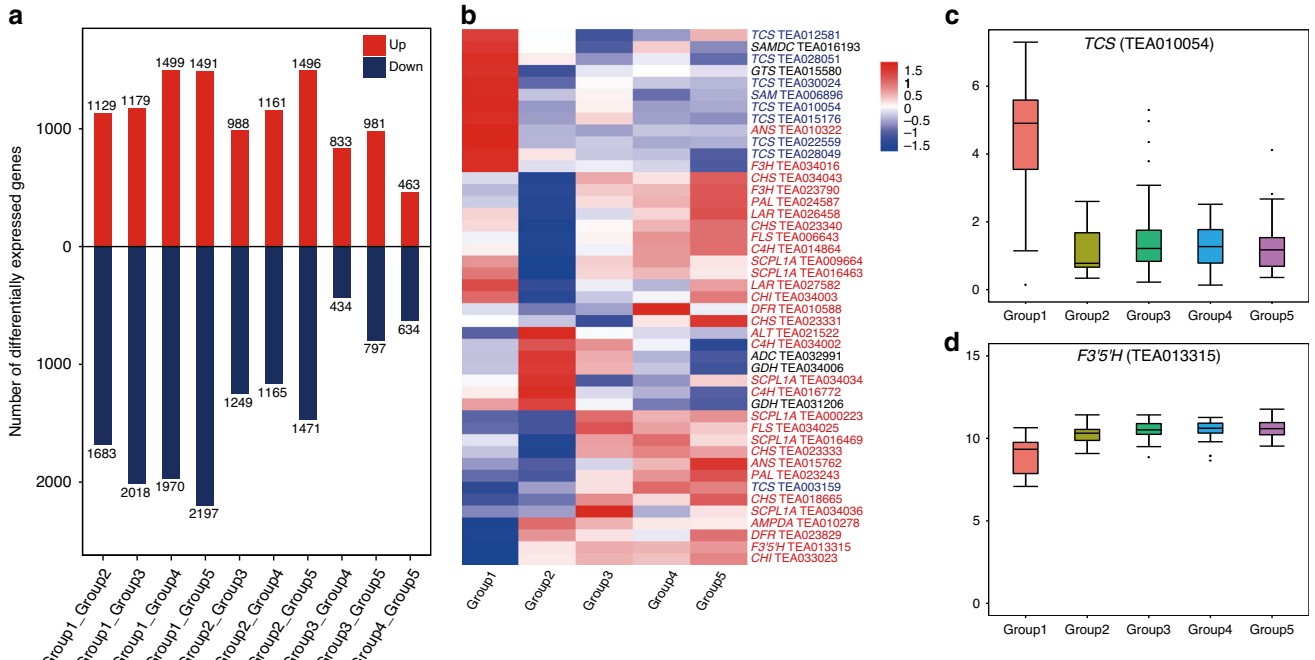

**Fig. 5 Differentially expressed genes identified in pairwise comparisons of five groups of tea accessions. a** Number of differentially expressed genes in pairwise comparisons of five groups of tea accessions. Numbers of up- and downregulated genes are indicated in red and blue, respectively. **b** Heatmap showing the expression patterns of genes that are known to be involved in the biosynthesis of catechins (in red), caffeine (in blue), and theanine (in black), and show significant changes in expression in at least one pairwise comparison. **c, d** Box plots of gene expression values (log2-transformed counts per million) of TCS and F3′5′H in different tea groups. (*n* = 25 for Group 1, 9 for Group 2, 28 for Group 3, 26 for Group 4, and 34 for Group 5). Boxes = interquartile ranges, middles = medians, whiskers = 1.5 × the interquartile range, single points = outliers. Source data underlying **b**, **c**, and d are provided as a Source Data file.

*TCS* (TEA010054), which encodes an *S*-adenosyl-ʟ-methionine (SAM)-dependent *N*-methyltransferase that catalyzes the methylation step to synthesize theobromine and caffeine, had a much higher expression level in group 1 than in other groups (Fig. 5c). However, no significant expression change was detected for the predominate *TCS* (TEA015791). TEA013315, with the highest expression level among all *F3′5′H* genes in tea plants, was expressed at the lowest level in the CSA lineage, and may have contributed to the apparent differential accumulations of catechin-derived metabolites between CSS and CSA (higher levels of C/EC/ECG in CSA, and a higher level of EGC in CSS) (Fig. 5d). However, the expression levels of genes related to caffeine or theanine biosynthesis did not show direct correlation with the concentration of respective metabolites (seven *TCS* were highly expressed in group 1, but group 1 did not have the highest caffeine concentration). Similarly, three genes involved in theanine synthesis, including arginine decarboxylase and two glutamate dehydrogenase genes, were highly expressed in group 2, yet group 2 did not have the highest theanine concentration (Figs. 4e, f and 5b), suggesting that much of the regulation of metabolite levels may occur post-transcriptionally.

## Discussion

The rich constituents of specialized metabolites in the growing tea leaf are believed to be essential for the flavor and quality of tea products[12,18,19,35,36,51]. Therefore, tea plant offers a good model to study the molecular and genetic basis underpinning the abundance, diversity, and regulation of specialized metabolites in plants. By analyzing transcriptomic and metabolomic data from 136 representative tea accessions in China, we were able to classify these accessions into five phylogenetic groups/populations, identify over 8000 polymorphic markers that can be used

for marker-assisted breeding, explore the dynamic variations in metabolite compositions and gene expression, and identify dozens of signature metabolites that are highly accumulated in one group of tea accessions but not in other groups. Our results show that there exists a high level of metabolite diversity in different tea populations and accessions, which can be explored to investigate the underlying regulatory mechanisms and guide molecular breeding for tea improvement.

With the transcriptomic data, we were able to identify 925,854 high-quality SNPs, providing a rich set of molecular markers that may be useful for marker-assisted breeding. Phylogenetic and population structure analyses showed that tea cultivars in China may be grouped into five populations, which is in general agreement with the results from previous studies using chloroplast DNA and nuclear microsatellite markers[34]. Our phylogenetic and population structure analyses showed that accessions from similar geographical origin, with similar morphological characteristics (e.g., large-leaved vs small/middle-leaved), or similar breeding/domestication history tended to cluster together nicely, as expected (Fig. 1b). With the genome-wide SNP markers that are mostly located within or near gene-encoding regions, our study not only provides reliable evaluation of genetic relationships and distances, but also offers a list of markers that may change the encoded protein sequences and markers with major alleles that are specific to certain tea groups, which are ideal markers for breeding practice. In a recent study, Xia et al.[35] used SNP markers genome-resequencing data separated 81 accessions into three clades (CSS, CSA, and wild tea), which disagrees with our conclusion or results using SSR markers[34]. This is likely because only a limited number of accessions (81) are analyzed, among which only 58 were cultivated accessions and thus the population did not fully represent the genetic diversity of natural tea populations. Another recent study used SNP markers from

transcriptome data to separate more than 212 accessions into five subpopulations[51], which is consistent with our phylogenetic analysis and results using SSR markers. However, many accessions were not grouped together by geographical origin or morphological characteristics such as leaf size and there was no distinction between wild tea accessions and cultivated accessions[51]. These results are not consistent with our data. This is likely caused by a small number of SNP markers, derived from a smaller amount of transcriptome data, being used for phylogenetic analysis[51].

Our untargeted metabolomics data show that thousands of metabolic features can be detected in fresh tea leaves, in addition to the well-known catechins, caffeine, and theanine. However, the majority of these detected metabolic features were lowly accumulated, with only 25–28% having a relative abundance higher than 500 in at least one examined accession. After careful annotation and manual curation of highly accumulated metabolic features (753 and 503), we found that 74% and 45% of them were fragment ions that were generated by the mass spectrum fragmentation process under POS and NEG mode, respectively, and thus were not natural metabolites in tea plants. After removing fragment ions, the identities for 179 and 258 abundantly detected tea metabolites could be confidently assigned or putatively assigned, representing the largest metabolite identification effort made in tea plants to date. While flavanols are the most abundantly occurring group of phenolic compounds in fresh tea leaves, flavonol glycosides and proanthocyanidins emerge as the most diverse ones. In particular, dozens of kaempferol and quercetin derivatives, with some not being reported in fresh tea leaves previously, are found to vary both in structures and concentrations across different groups of tea accessions. It is increasingly recognized that tailoring enzymes such as those catalyzing glycosylation, acylation, and methylation make a greater contribution to the structural variations of flavonoid metabolites. Their structural diversities in turn determine their biological activities, which are often implied in conferring tolerance to various stresses[52,53]. The natural variations of flavonol glycosides in tea leaves, many of which are observed to be heavily decorated by various sugars as well as coumaric acid, are presumed to be ascribed to the differential activities of specific UDP-dependent glycosyltransferases and acyltransferases yet to be functionally characterized[52,54]. Unraveling candidate genes responsible for flavonol decoration and teasing out which enzymes are functionally important will shed light on flavonoid biosynthesis in tea plants.

Our comparisons of metabolites from different tea groups suggest that the CSA tea type has a distinct metabolite profile from that of CSS tea type, resulted from natural and/or agronomic selection. It is well-known that tea plants contain high level of catechins, among which EGCG and EGC have the highest accumulation, followed by ECG, EC, EC, and C[20,55,56]. Our comparison of metabolite contents in different groups of tea plants show that the CSA tea accessions have higher accumulation of diverse classes of flavonoids (e.g., C, EC, GC, ECG, flavanols, flavonol glycosides and procyanidin dimers) and derivatives of gallic acid and quinic acid, with relatively lower level of EGCG. These results are in agreement with a previous targeted metabolomic analysis of catechin contents in 403 Chinese tea germplasms[50]. Green tea accessions contain lower levels of catechin compounds and relatively higher levels of two triterpenoid saponins and galactosylated derivatives of kaempferol/quercetin glycosides. During green tea processing, major catechins from young leaves of *C. sinensis* remain unoxidized. It is generally believed that a lower ratio of total polyphenols to amino acids in fresh leaves is essential to balance the astringent and the mellow tastes, and hence a prerequisite for producing premium

green teas[57,58]. Oolong tea accessions are enriched with two methylated catechins and complex kaempferol/quercetin glycoside derivatives acylated with a coumaroyl group. Interestingly, the study by Lv et al.[59] also suggests that oolong tea cultivars may be a good source for finding tea cultivars with higher methylated catechins. The extensive variations of catechins and some other metabolites that were revealed by this study suggest that metabolic profiles may be used to distinguish tea cultivars and metabolic markers may be used to assist tea breeding. On the other hand, environmental factors are known to greatly affect the accumulation levels of specialized metabolites. Future studies should determine whether signature metabolites are quantitatively affected by a specific environmental factor, as this information may help to determine growth conditions that optimize the production of a desired metabolite.

What are the underlying molecular mechanisms for the apparent differential accumulations of catechin compounds in different tea groups? Although we detected 31 structural genes in the catechin biosynthesis pathway that were differentially expressed in different tea groups (Fig. 5b), direct correlation between gene expression level and metabolite level is not obvious. For example, the anthocyanin reductase (ANR) is responsible for converting delphinidin to EGC and cyanidin to EC, but the ANR gene is not expressed at a higher level in CSA than in CSS tea accessions. Additionally, many structural genes have multiple copies in the genome and different copies in the same gene family may display different expression patterns. For example, the anthocyanidin synthase (ANS) is responsible for converting leucocyanidin to cyanidin and leucodelphindin to delphinidin and we found that two ANS genes (TEA010322 and TEA015762) displayed opposite expression patterns with TEA010322 being highly expressed in CSA tea accessions and TEA015762 being highly expressed in green tea and oolong tea accessions (Fig. 5b). Similar pattern was also observed for the two LAR genes (TEA026458 and TEA027582) that encode the leucoanthocyanidin reductases that are responsible for converting leucocyanidin to C and leucodelphindin to GC. On the other hand, caffeine did not show significant change in accumulation levels in different tea groups (Fig. 4f), indicating that it is an integral part of metabolites for any tea products and traditional breeding by crossing different tea cultivars is not effective in changing their concentration. Nevertheless, we found that nine genes in the caffeine biosynthetic pathways that were differentially expressed (Fig. 5b), again suggesting no direct correlation between gene expression and metabolite level. These results suggest that much of the regulation of metabolite levels may not occur at the transcriptional level. Further studies are needed, probably through genome-wide correlation analyses between metabolite concentrations and gene expression levels or molecular markers, to identify key regulators for metabolite production[9].

## Methods

**Sample collection**. We collected the fully expanded second leaves from the young shoots (one bud with two leaves) of 136 representative tea accessions (belonging to 128 cultivars) grown in major tea-growing regions (e.g., Fujian, Zhejiang, and Yunnan Provinces) in China from April 13th to 25th, 2018 (Fig. 1 and Supplementary Table 1). For each tea accession, three biological replicates were prepared for RNA-sequencing and five biological replicates were prepared for metabolomics analysis with each replicate representing a pool of leaf samples collected from 15-20 individual tea plants of the same accession. Fresh tea leaves were immediately frozen in liquid nitrogen, brought back to the laboratory and stored at −80 °C until further analysis.

**RNA-sequencing and RNA-seq data analysis**. Total RNA was extracted using the CTAB (BBI Life Sciences, Shanghai, China) and PBIOZOL (Bioer, Hangzhou, China) reagents according to the manufacturer's protocol. RNA concentration and integrity were examined with the Agilent Bioanalyzer 2100 system (Agilent, CA, USA). Oligo (dT) beads were used to isolate poly(A)-containing mRNAs, which

were fragmented into ~250 bp fragments. cDNA libraries were constructed according to the standard protocol from Beijing Genomics Institute (Shenzhen, China) and paired-end 100 bp reads were generated on a BGISEQ-500 platform with a depth of approximately 5 GB clean data per sample. Transcriptomes from four wild relatives of tea plants in genus *Camellia*, including *C. japonica*[60], *C. azalea*[61] *C. nitidissima*[62], and *C. reticulata*[63] were downloaded from the Gene Expression Omnibus (GEO) database (https://www.ncbi.nlm.nih.gov/geo/) to be used in the current study.

Raw RNA-seq reads were processed with SOAPnuke[64] to remove low-quality regions and adapter sequences. Clean reads were mapped to the CSS reference genome (downloaded from http://pcsb.ahau.edu.cn:8080/CSS) using hisat2[65] and gene expression levels were summarized by HTseq-count[66]. Raw counts were then normalized to counts per million (CPM) and genes with CPM < 1 in 90% samples were regarded as lowly-expressed genes and were removed from further analysis. Normalized gene expression was log2-transformed and used for clustering analysis with *t*-SNE in R version 3.5.1. Differentially expressed genes among five groups of tea varieties were identified by performing pairwise comparisons using edgeR[67] with significance thresholds of false discovery rate (FDR) < 0.05 and fold-change >2. Approximately 40–120 million reads for each sample were uniquely mapped to the reference genome. To minimize the effect of library size on quantification of genome expression patterns, the total uniquely mapped reads larger than 80 million were then down-sampled to 80 million reads with GATK v4.0.4.0[68].

**Evolutionary analyses**. To identify SNPs among the collected tea varieties, clean reads were further processed to filter PCR duplicates, and retained reads were used to call variants following the mapping process with GATK v4.0.4.0[68]. The HaplotypeCaller function was then used to generate a GVCF file for each accession, followed by population variant calling with the function GenotypeGVCFs. Hard filtering was applied to the raw variant set using GATK, with parameters "QD < 2.0| | FS > 60.0| | MQ < 60.0| | MQRankSum < −12.5| | ReadPosRankSum < −8.0" to obtain high-quality SNPs. Redundant SNPs were discarded such that candidate SNP loci were more than 5 bp away from each other. Only biallelic SNPs with a minor allele frequency larger than 0.05 and missing rate less than 20% in all samples were retained as final candidate SNPs for further analysis. Candidate SNPs in coding regions were further classified into synonymous SNPs and non-synonymous SNPs with ANNOVAR[69]. A major allele for an SNP in each tea group is defined as the allele with a frequency of at least 0.75 in the group.

Only non-missing SNPs at the fourfold-degenerate sites were selected to estimate genetic distances across all samples. An approximate maximum likelihood phylogenetic tree was constructed with 45,162 fourfold-degenerate SNPs using FastTree[70] with 1000 bootstrap replications. The wild tea (S159), together with the aforementioned wild relatives of tea plants, was used as the outgroups for rooting the tree.

The genetic relationship of 134 accessions was estimated using PCA performed by using PLINK v1.9[71]. Population structure was inferred with STRUCTURE[44]. To determine the optimal number of populations, STRUCTURE was run 10 times and with 20,000 MCMC reps for each *K* (*K* = 2–9). The optimal *K* was estimated to be 5 with Harvester[44].

Based on the high-quality SNPs identified in tea accessions, selective sweep regions were detected among five groups of tea varieties with XP-CLR[45]. XP-CLR estimated each scaffold in non-overlapping 10-kb windows with a 10-kb sliding-step to detect allele frequency differentiation between each two populations across each reference genome region. Adjacent windows with the highest XP-CLR scores (5%) were grouped into a single region and regions with the top 1% XP-CLR scores were considered as potentially selected sweeps. Nucleotide divergence ($\pi$) in each group was also calculated in 10-kb sliding windows with 1-kb steps across the reference genome which aided in improving prediction accuracy. Only potential selective sweep regions that were identified by XP-CLR and had a top 50% $\pi$ ratio were kept as candidate sweeps.

**Metabolomics analysis**. Metabolite extracts were prepared by adding 750 μL of 70% methanol to 30 mg (±0.5 mg) of the ground and pre-lyophilized leaf samples as previously described[20]. The samples were spiked with 250 μL of 200 μg/m 2',7'-dichlorofluorescein as an internal standard. A 10 μL aliquot was further diluted 100-fold with 70% methanol and filtered through a 0.22 μm polyvinylidene fluoride (PVDF) filter (Millipore, Billerica, MA, USA).

Data acquisitions were performed using an LC-MS system, which is a Waters Acquity UPLC system coupled in tandem to a Waters photodiode array (PDA) detector and a SYNAPT G2-Si HDMS QTOF mass spectrometer (Waters, Manchester, UK). Gradient elution was achieved on a Waters Acquity UPLC HSS T3 column (100 × 2.1 mm, 1.8 μm) with water containing 0.1% formic acid (solvent A) and acetonitrile containing 0.1% formic acid (solvent B) at a flow rate of 0.3 mL/min. The column temperature was maintained at 40 °C. The gradient elution program was as follows: 1–7% B (0–2 min), 7–40% B (2–13 min), 40–60% B (13–17 min), immediately elevated to 99% B (17 min), held at 99% B (17–22 min) and allowed to equilibrate for a further 3 min prior to the next injection. The last 8 min of the chromatogram solutions were discarded. The injection volume was 1 μL. MS data were recorded using a QTOF mass spectrometer with an ESI source and operated in both the positive and the negative modes. The MS data were acquired in continuum mode using ramp collision energy from 10 to 50 eV. The following

MS parameters were applied: capillary voltage, 2.5 kV (ESI⁺) and 2.0 kV (ESI⁻); cone voltage, 40 eV; collision energy, 4 eV; source temperature, 120 °C; desolvation temperature, 450 °C; cone gas flow, 50 L/h; desolvation gas flow, 800 L/h; *m/z* range, 50–1200 Da. Quality control (QC) samples were prepared by pooling the equal amount of all second leaf samples and were injected every ten samples throughout the analytical run to check instrument performance. The instrument was operated under the control of the MassLynx software (ver 4.1, Waters, Milford, MA, USA).

Components eluting between 1 and 17 min from the UPLC-QTOF MS system were processed in Progenesis QI (v2.1, Nonlinear Dynamics, Newcastle upon Tyne, UK) for data preprocessing with default settings, except that each sample was normalized to the internal standard. Subsequent multivariate analyses, such as principal component analysis (PCA) and partial least squares discriminant analysis (PLS-DA), were carried out by Progenesis QI extension EZinfo, following Pareto scaling. After manual inspection to remove outliers, the datasets including mass features and normalized peak area (relative abundance) were exported to Microsoft Office Excel for subsequent statistical analysis. Compound information obtained from Progenesis QI was used as the start point for manual metabolite identification. First, metabolites, where authentic standards were available, were verified by comparisons of their retention time and MS/MS fragmentations. When no authentic standards were found, tentative identification was made by comparing the mass spectra with those in online spectral databases of Metlin[72], MassBank[73], HMDB[74], KNApSAcK[75], and ReSpect[76] and verified with the literature information on similar compounds, especially for those that had been reported in tea. Collision-induced dissociation (CID) fragmentation of selected ions, if needed, was performed to confirm the structural assignment. UV spectra were used for identification whenever possible.

Outlier metabolite data were detected and discarded based on the median absolute deviation (MAD) method in the five replicates of each sample. Metabolites with relative abundance <500 in all samples were regarded as lowly accumulated metabolites and were removed from further analyses. One-tailed Student's *t*-test was performed to identify differentially accumulated metabolites (DAMs) and the Benjamini–Hochberg (BH) correction was used to adjust *p*-values due to multiple comparisons. The metabolites with an adjusted *p*-value less than 0.05 and fold-change larger than 2 were regarded as DAMs. Using the total panel of metabolite values as the reference control, all data were normalized and then log2-transformed for *t*-SNE clustering analysis in R version 3.5.1.

**Reporting summary**. Further information on research design is available in the Nature Research Reporting Summary linked to this article.

## Data availability
RNA-sequencing data that support the findings of this study have been deposited to the Gene Expression Omnibus (GEO) database at the National Center for Biotechnology Information (NCBI) and are accessible with project number "PRJNA562973". Metabolomics data have been deposited to the MetaboLights database[77] at the EMBL-European Bioinformatics Institute (EBI) with project number "MTBLS1405". All other relevant data are available from the corresponding author on request. Source data are provided with this paper.

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

## Acknowledgements

This work was supported by the Fujian Agriculture and Forestry University (FAFU) Construction Project for Technological Innovation and Service System of Tea Industry Chain (K1520005A02) and other funds from FAFU to X.Y., Y.Y. and R.L. We wish to thank Dr. Keke Chen, Dr. Changsong Chen, Shixian Chao, and Yinbi Cai for their assistance in collecting tea samples.

## Author contributions

Z.Y. and R.L. designed and coordinated the study. X.Y., Y.Y., S.C., J.M., J.L., Z.F., Q.Z., Q.C., and L.C. collected the tea leaf samples. X.Y., S.C., Y.L., and R.L. performed the metabolomics analyses. J.X., X.Y., S.C., and R.L. analyzed the data. R.L., J.X., X.Y., and Y.Y. wrote the manuscript with input from all authors. R.L., L.C., and Z.Y. edited the manuscript. All authors read and approved the final version of the manuscript.

## Competing interests

The authors declare no competing interests.
