## [Peer Review File · Nature Communications]

Reviewers' comments:

Reviewer #1 (Remarks to the Author):

The manuscript entitled as "Metabolite signatures of diverse *Camellia sinensis* tea populations" by Yu et al. described an integrated metabolite and transcriptomic profiling on about 136 *Camellia sinensis* plants, representatives of tea populations from most major tea-production provinces. By identification of less than 400 defined metabolites and many other unknown ion peaks, representing metabolic features of tea plants (no more than 4000), and about 930 K SNPs, among them, about 1100 SNPs were identified as differential sites for CSA and CSS varieties.

Through SNP based clustering, 5 groups of these tea populations are thought to feature distinguishingly from each other. Both SNP and metabolite features in these 5 groups (one CSA group and five CSS subgroups) of tea populations were analyzed, authors thus claimed that their studies gain new insight into evolution, adaptation, and domestication of tea plants.

Generally, this manuscript provided many confirmative information to our current understanding of tea plant populations, evolutionary relations, and some metabolite production characteristics. Various metabolite signatures were also reported by scattered studies in JAF, BMC series journals, Food Chemistry, or other journals often publishing tea plant researches.

In recent years there are already many similar studies on metabolites and transcriptome analysis have been done, either on small samples or large samples (13, near 53), or different genotyping on tea plant natural or artificial populations (from 20 to more than 400 varieties), have been reported (Food Chemistry, JAF, BMC genomics, 2019; etc). For example, a recent study using 450 tea plant populations for 2b-RAD genome sequencing have revealed more SNP and tea population structure (BMC Genomics, 2019).

This work is only based on relatively larger representative tea plant varieties, compared with previous ones, but without really done metabolite-gene expression association, may not give more advanced scientific finding or insight. As revealed by studies on other crops, such as rice and maize, soybean, rapeseed, etc, the 136 population size or representative tea plant varieties are too small to give any substantial or solid finding, particularly on tea plants, with genome size of ~3.1 Gb, and so complex secondary metabolism.

Indeed, these SNP analysis supported the information about different origins of these tea plant cultivars that are planted in these provinces, author confirmed and discussed the relationships between each other groups, which are actually mostly known.

So this study may be too preliminary to be regarded as an advanced understanding or important progress made in tea plant researches.

There are also some questions need to be clarified

1. Authors only sequenced each tea plant variety for only 5G clean data, which is less than 2 X deep sequencing. it is very not enough for a complex genome with more than 36000 genes annotated, not including many transcript unidentified or annotated.

Under these data size, one may not be able to get SNPs reliable enough, so I doubt about the 930K SNPs are really high-quality?

2. Since the population study is set for tea plant evolution, author have not used several other plants in Tea section, and wild tea trees that could help distinguish sub-species, such as CSA and CSS.

3. This study confirmed the common observation, CSA- also known large-leaf tea plants, and CSS- also known as small leaf tea plants, and some middle-leaf tea plant-as either adaptive to warm or cold environments or hybrids of CSA-CSS. CSA plants had more flavonols, EGCG catechins, and hydrolysable tannins, as compared with CSS. but this study did not gid out their specific reasons for that, even certainly, phenylpropanoid genes most likely are more highly expressed in CSA than in CSS, as indicated by previous studies.

4. Although it is well-known that the accumulation of plant specialized metabolites, including tea plant characteristic metabolites, such as catechins, theanine, caffeine, and various volatiles, mostly are results from plants suffering various abiotic and biotic stresses, and there are highly variable relationships between certain metabolites and their corresponding synthetic genes or regulation factor genes, it is still possible to draw some reliable and solid relations between key gene expression with these metabolites. Authors should step further to draw the relationships, by using more other data, either public or their own experimental data.

5. Authors somewhat had mistaken these tea plant genetic variations, tea processing suitability, with local people 's traditional tea processing favorites. Because green teas can be made in all provinces with the tea plant leaves harvested from their local tea plant varieties. so do other types of teas, such as Oolong. The famous types of Oolong teas in Fujian mostly is due to their sophisticated Oolong tea processing techniques, their marketing results, or other non-plant scientific factors. This may not be due to their specific metabolites, or genetic backgrounds. The Oolong teas now are also produced by other provinces by using local tea leaves with similar quality proved that points.

6. Authors identified some evolutionary selection sweep regions, among more than 800 selection sweep regions, two of them contained caffeine synthetic gene AMPDA, and tri-hydroxylated catechins, these may suggest caffeine and catechins are evolved at least through these sweeps. However, metabolite profiling indicated that among 136 tea varieties, catechins changed significantly, in both quantities or quality. but caffeine is almost similar in content, similar to theanine. Why these controversial results were obtained?

7. While the title of the manuscript is about metabolite signature of tea *Camellia sinensis* population, the metabolite profiling, identification, and classification were far behind these of transcriptome analysis.

Tea plants contain numerous secondary metabolites with diverse structures and physiochemical properties, it is a big challenge for most labs to properly profiling and precise identification of these tea plant specialized metabolites, as these done in rice and maize, it has to be realized that no enough number of identified specialized metabolites certainly limited the in-depth understanding of tea plants, since these are their most significant features.

Particularly, these differentially accumulated metabolites between CSA and CSS, as well as among CSS subgroups. However, only 57 annotated metabolites among 355 positive ion peaks, or 76 metabolites with identified names among 286 negative ion peaks. As tea plants are featured with their diverse and complex metabolites, more annotation work on these metabolites should help to illustrate more or real signature metabolites, among these tea populations.

8. The confirmative identification of metabolite features or commonly observed metabolite phenotypes on these tea plant varieties in this study limited the capability of this study to look really insight in the substantial features. The limitation is most likely due to these unreachable or not unveiled metabolites in these tea plant samples or too small size of total tea plant samples.

Reviewer #2 (Remarks to the Author):

The manuscript “Metabolite signatures of diverse *Camellia sinensis* tea populations” by Yu and coworkers provides an interesting and very useful analysis of the metabolites found in a collection of tea germplasm, grown under field conditions at various locations in China. The research is comprehensive and appropriately performed. Some of the interpretations could be improved, but this only requires a rigorous rewrite. I provide a few suggestions, in page order, below.

On two general points, the authors should emphasize that the metabolites they analyze are from unprocessed leaves, and are expected to be significantly different from those that would be seen in the processed teas that are actually consumed. I believe that there is quite a bit of literature on this subject, and some of it should be referenced in both the Introduction and the Discussion. The authors also need to tone down their conclusions vis-à-vis the significance of their transcriptome results (see Page 14 comments below) and of their selective sweep results (see Page 9&10 comments below)

Page 2

It is not clear whether the “abundance” refers to metabolites or tea plants. I suggest that the first sentence of the abstract be changed to “...system to study the evolution and diversification of the numerous classes, types and variable contents of specialized metabolites.”

I suggest “which are beneficial for targeted tea breeding and improvement” be deleted. This is a generic sort of statement, and the point was already made on lines 34, 35 and 37 in the Abstract.

Page 3

Line 71: “plant specialized” should be “plant-specialized” because it is a compound adjective.

Line 77: “validations” should be “validation”

Line 78: “at an early stage to answer” is awkward wording, with “cannot yet answer” being a superior replacement

Line 85: “Tea plant” should be “The tea plant”

Page 4

Line 88: “that offers” should be “and offers”

Line 88,89: “anti-cardiovascular and anti-allergic activities” should be “anti-cardiovascular disease and anti-allergy activities”

Line 100: Actually, I believe that theanine is produced in many plants, and even some fungi, and is not unique to tea. It is particularly abundant in tea.

Line 107: It is not proven that China is the place where tea originated, although the area between Assam, Yunnan and Burma seems most likely at this time. It is clear that tea was first officially domesticated in China. By the way, the correct phrasing is “center of origin” not “origin center”

Page 6

Line 155: “natural variations” should be “natural variation”, while “and maintained and propagated” should be “and are mostly propagated”

Line 174: To what does “wild tea” correspond? Wild tea should mean *Camellia sinensis* that was not planted or otherwise maintained by humans. Are you calling these other *Camellia* species “wild tea”? If so, that is incorrect and should be replaced by “tea relatives” or “close tea relatives”

Line 177: “comprising exclusively of CSA type accessions” should be “containing exclusively CSA accessions”

Page 7

Line 182: What does “were adopted from wild tea plants” actually mean? Do the authors mean “adopted” or “adapted”? All domesticated tea originally was adapted from wild tea plants, so it is unclear what is meant by these authors in the case presented here.

Line 196: “using” should be “using the”

Line 199: “from the crossing between” should be “from a cross between”

Page 9

Line 243: “sweep” should be “sweeps”

Line 255: “by natural environment” should be “by natural environments”

Lines 262-269: Regarding the candidate selective sweep regions, is there any enrichment for genes that are candidates for known tea quality metabolites? Of course, with 833 regions, you will find some with genes involved in tea leaf metabolites just by chance, but is there any enrichment? Depending on how one sets the criteria, potential selective sweeps can always be “found” but that does not mean they are anything other than chance regions of low polymorphism

Line 270: “indicated” is way too strong, and would be better replaced by “suggested”

Page 10

Line 272: as indicated by my comments for lines 262-270, I think that “were the” should be replaced by “may have been”

Lines 281-284: If, as the M&M says, these leaves were fast frozen (as in ref 20), then this needs to be stated somewhere here. This was not a metabolite analysis of processed tea, but rather of tea leaves.

Lines 295-300: What was the statistical significance analysis on this? A multiple testing correction (e.g., Bonferroni) would mean that differences would need to be very high and very consistent for any of these results to be established as significant.

Page 11

Line 323: “spectroscopy” should be “spectroscopic”

Page 12

Line 344: “was featured with” should be replaced by “consistently exhibited”

Line 361: “twice of that” should be “twice that” while “accessions” should be “accessions,”

Page 14

Lines 402-418: The authors make much of the differences observed at the transcription level for these genes, but page 11 results indicated that they did not vary much at the level of the metabolites produced. This point is then made, very appropriately, by the authors on lines 418-422. However, just saying this is “rather complex” is too little. I think the Discussion needs a place where it is stated that these results indicate that much of the regulation of metabolite levels occurs post-transcriptionally, and thus it is not clear how often (or if ever!) the transcriptional results may be informative relative to significant differences between the cultivars.

Page 15

Lines 425-426: The first sentence is generic and obvious, so it should be deleted. The second sentence is not good English and somewhat begs the question of the entire study, so “The rich constitutes of specialized metabolites are essential” should be replaced by “The rich constitution of special metabolites in the growing tea leaf are believed to be essential”

Line 432: “polymorphism” should be replaced by “polymorphic”

Page 16

Lines 460-461: Are somewhat obvious statements, poorly worded and not necessarily true. That is, how long has the divergence of CSS and CSA types been ongoing? If it is just a few thousand years, then that is unlikely to be enough time for significant random drift/divergence. I would suggest that the lines “resulted from long-term divergence and natural or artificial selections” be replaced by “resulting from natural and/or agronomic selection”

Line 462: “are featured with” should be replaced by “have”

Lines 465-467: I expect this is true, but where are the references to support these ideas? If you cannot reference the ideas, then this just be stated as a possibility, not as a certainty.

Page 17

Line 504: I believe the authors mean “reticulata” not “reticulate” and “from the public database” is not sufficient unless the authors tell us which public database(s) they are referring to.

Page 18

Lines 523-533: It would be good to provide more information on how heterozygosity was called, and also to have a supplementary table that describes overall heterozygosity for each accession investigated

Line 543: “run for 10” should be “run 10”

Page 19

Line 547: “with 10-kb” should be “with a 10-kb”

Line 552: The authors show some inconsistency in the manuscript in how they report DNA fragment sizes. On this line, they have written “10-kb” and “1kb”. They need to decide on whether they will use the “-” always or never in this manuscript for this type of designation, and then be consistent throughout.

Lines 553-554: This statistical analysis is key, but most readers will not be knowledgeable in this pi value approach, nor am I clear on the degree to which it takes into account multiple testing nor in how it relates to a statistical value most readers will understand, like a p-value.

Reviewer #3 (Remarks to the Author):

The manuscript entitled “Metabolite signatures of diverse *Camellia sinensis* tea populations” characterized the population structure and phylogenetic relationships among major tea cultivars and association of tea metabolites with populations using transcriptomic and metabolomic analyses. This is a huge work and obtained significant findings, which will contribute to future targeted tea breeding and improvement. The following points are suggested to be improved.

(1) The abstract need be revised. The precise highlighted findings should be indicated.

(2) Based on the results, the authors suggest that PAL and TCS in different groups contain SNP sites, but further experiments are needed to clarify the functional differences of these SNP sites for PAL and TCS in order to rule out the possibility of synonymous SNPs. Whether or not these SNPs affect the differences in enzyme activity or other properties of the two enzymes, thus affecting the differences in the contents of certain metabolites in different groups.

(3) The authors suggest that the regions of AMPDA and F3'5'H may be selected, but the corresponding downstream metabolites need to be determined in different populations to show that the two genes are indeed selected in different populations.

(4) Can the characterization of metabolites be analyzed in association with the SNP previously measured? Possible correlations during the period of speculation.

(5) What are the possible down-regulation reasons for TEA016601 (FLS), TEA023333 (CHS) and TEA023790 (F3'H) in S164 sample?

Point-by-Point responses

Dear Reviewers:

We thank all three reviewers for your constructive comments and critiques that have helped to improve our manuscript. In the following we provide our responses (in blue color text) to your comments (black text) and highlight how we have addressed your concerns. **Please note that major changes in the main text are highlighted in yellow.**

Sincerely yours,

Renyi Liu
Zhenbiao Yang
Liang Chen

Reviewer #1:

"The manuscript entitled as "Metabolite signatures of diverse *Camellia sinensis* tea populations" by Yu et al. described an integrated metabolite and transcriptomic profiling on about 136 *Camellia sinensis* plants, representatives of tea populations from most major tea-production provinces. By identification of less than 400 defined metabolites and many other unknown ion peaks, representing metabolic features of tea plants (no more than 4000), and about 930 K SNPs, among them, about 1100 SNP were identified as differential sites for CSA and CSS varieties.

Through SNP based clustering, 5 groups of these tea populations are thought to feature distinguishingly from each other. Both SNP and metabolite features in these 5 groups (one CSA group and five CSS subgroups) of tea populations were analyzed, authors thus claimed that their studies gain new insight into evolution, adaptation, and domestication of tea plants.

Generally, this manuscript provided many confirmative information to our current understanding of tea plant populations, evolutionary relations, and some metabolite production characteristics. Various metabolites signatures were also reported by scattered studies in JAFc, BMC series journals, Food Chemistry, or other journals often publishing tea plant researches.

In recent years there are already many similar studies on metabolites and transcriptome analysis have been done, either on small samples or large samples (13, near 53), or different genotyping on tea plant natural or artificial populations (from 20 to more than 400 varieties), have been reported (Food Chemistry, JAFc, BMC genomics, 2019; etc). For example, a recent study using 450 tea plant

populations for 2b-RAD genome sequencing have revealed more SNP and tea population structure (BMC Genomics, 2019)."

Response: We agree with the reviewer that there were previous studies focusing on the population structure or metabolite contents of tea populations, however, with due respect, we cannot agree that our results are merely "confirmative" but believe that our findings are novel and significant for the following reasons:

(1) Our study has revealed major signature metabolites for each of the five tea phylogenetic populations. This is an **important novel finding** that will be of great interest to tea scientists and plant scientists and will be of great value for tea improvement and molecular breeding of new tea cultivars.

(2) **Our comprehensive untargeted metabolomic study of fresh leaves from 136 widely distributed tea accessions is unprecedented**, and has identified many metabolites that have not been reported in tea plants before (see below. response to points #3 and #7 of reviewer #1), and **is clearly distinct from all previous tea metabolite analyses**. **First of all**, most previous metabolomic analyses of large tea samples were carried out on processed tea products (i.e. products that are ready for human consumption)^{1, 2, 3, 4}, rather than fresh leaves. It is well-known that metabolite contents change dramatically after fresh tea leaves are processed into tea products, making it difficult to correlate metabolic profiles to genetic backgrounds. **Furthermore**, no previous untargeted metabolomic analyses cover the diverse tea populations and geographic distributions, as we did in our study. There were a few studies using fresh tea plant tissues for metabolic profiling, but they are limited by the genetic diversity and sample size of the population studied. For examples, our previous metabolic profiling of 14 Wuyi Rock tea cultivars through UPLC-QTOF MS-based untargeted metabolomics revealed some key metabolites, such as catechins (EGCG, ECG, EGC, EC, GC and EGCG3"Me) and flavonol glycosides (rutin, quercetin galactosyl rutinoid, quercetin glucosyl rutinoid, kaempferol rutinoid and kaempferol glucosyl rutinoid) to be important for cultivar discrimination⁵. Li *et al.* profiled the metabolites in three albino tea cultivars and two green tea cultivars by GC-MS and LC-MS and found that amino acid metabolism and sugar metabolism were altered between two genotypes, in addition to several flavonoid compounds (*e.g.*, catechins, proanthocyanidins and anthocyanins)⁶. However, we could not obtain a clear picture of signature metabolites of diverse natural tea populations from these limited analyses, which is what our current study aims to provide. **Finally**, our tea samples were collected from a wide range of geographic locations in China, and thus the metabolic signatures we identified for each tea phylogenetic clade are independent of geographic and environmental effects, assuring that the consensus signature metabolites only result from genetic effects.

(3) *Our genome-wide high-quality SNP analyses of tea population structure and phylogenetic relationship involves the largest and most diverse tea populations and deepest genome-wide sequencing data, generating the most clear-cut phylogenetic*

groupings of tea populations with highest confidence to date. Previous genetic studies of tea populations did not use genome-wide markers (mainly used a small number of molecular markers such as SSRs or chloroplast markers, number of markers ranged from 14 to 118)^{7, 8, 9, 10, 11, 12, 13, 14}, or sampled a limited number of tea plants (e.g. 18¹⁵, 40¹⁶ and 48¹⁷), or only investigated populations that were distributed in a narrow geographical region (for example, in Qinba area¹⁴ and Guizhou plateau in China¹⁸ or in Uji and Shizuoka areas in Japan¹⁹). A few recent studies used artificial populations (offspring from cross of two known parental plants) for the discovery of loci related to certain important traits such as timing of spring bud flush, catechin content, or flavonoid content^{20, 21}. These artificial populations are completely different from the natural population that our study focused on. In the study cited by the reviewer (Xu et al. BMC Genomics 2018), the authors used a reduced sequencing technology (2b-RAD sequencing) to obtain polymorphic markers on a F1 segregating population consisted of 327 individuals with the purpose of finding QTLs related to flavonoid or caffeine content. Only 13,446 SNP markers were discovered and only 4,463 of these markers were available for constructing the genetic linkage map. This artificial population came from a cross between two parental tea cultivars: Longjing 43 and Baihaozao²². Therefore, this study has essentially no overlap with our work in terms of the questions studied, experimental design, and materials/population used.

When we were revising our manuscript, two articles were published with some analysis of natural tea populations. By using genome resequencing data, a study published online in April 2020 divides tea accessions into three clusters (CSS, CSA, and wild type), most likely due to limited genetic diversity of the 81 tea accessions resequenced²³. However, our results show the existence of five major tea clades (which agrees with previous results by Yao et al. using SSR markers²⁴), based on genome-wide markers (925,854 high-quality SNPs) from 136 accessions representing genetically diverse tea populations. Another study published on July 24th, 2020 analyzed the population structure of 212 accessions using SNP markers from transcriptome data²⁵. Although they found five subpopulations, which agrees with phylogenetic analysis results from our data and previous SSR data, they found that these accessions were not grouped together by geographical origin or morphological characteristics such as leaf size and that there was no distinction between wild tea accessions and cultivated accessions (different from our results and results from genome-resequencing data or SSR marker data). This is likely caused by a small number of SNP markers that they used to build phylogenetic tree (because they collected a much smaller amount of RNA-seq data for each accession: 215 MB to ~14GB with a median 7.8 GB). Most importantly, our population structure analysis, integrated with untargeted metabolomic studies, showed that these five clades can be clearly separated not only by genome-wide markers but also by signature metabolites.

(4) The samples used in our study were collected from tea accessions that were grown in field conditions, and thus their metabolic and transcriptomic profiles represent real-time information about tea leaves before they were processed into

commercial tea products and reflect the interaction between genetic background and environment. To the best of our knowledge, our project is the first to use such design in the study of natural tea populations.

"This work is only based on relatively larger representative tea plant varieties, compared with previous ones, but without really done metabolite-gene expression association, may not give more advanced scientific finding or insight. As revealed by studies on other crops, such as rice and maize, soybean, rapeseed, etc, the 136 population size or representative tea plant varieties are too small to give any substantial or solid finding, particularly on tea plants, with genome size of ~3.1 Gb, and so complex secondary metabolism."

Response: The primary goal of this study is to identify metabolite signatures of different tea phylogenetic groups. Metabolite-gene expression association study is out of the scope of this manuscript.

"Indeed, these SNP analysis supported the information about different origins of these tea plant cultivars that are planted in these provinces, author confirmed and discussed the relationships between each other groups, which are actually mostly known.

So this study may be too preliminary to be regarded as an advanced understanding or important progress made in tea plant researches."

Response: We believe that our results from the analysis of a large set of genome-wide SNPs not only conclusively established the tea population structures (e.g. tea populations can be classified into five major groups) implicated from previous preliminary results based on molecular markers. Our study encompassed 925,854 genome-wide SNP markers and a large number of representative accessions (136), giving high confidence of the evolutionary analyses, in contrast to the previous preliminary studies involving a smaller number of molecular markers, smaller population size, or more limited geographical distribution of the samples^{7, 8, 9, 10, 11, 12, 13, 14, 15, 16, 17, 18, 19} (please see our response #1 for more details). Furthermore, our analyses using data from five relatives of tea and the genome-wide SNPs resolved the tea phylogenetic relationship with high confidence. Importantly, we have identified 8,187 SNP markers that possess different major alleles in different tea groups and have discovered 1,132 selective sweep regions that contain 833 genes. Our work has firmly established tea population structure and phylogeny and laid a solid foundation for the GWAS-based functional studies in tea plants.

More importantly, as discussed above we performed untargeted metabolomics analysis of fresh leaves from these representative accessions (grown in field condition), compared the metabolite contents in different accessions and tea groups, and obtained signature metabolites in different tea groups. Taken together, our results not only give us a clear picture of phylogenetic relationships and population structure of natural tea populations, but also provide new information on molecular

markers, metabolite compositions, and gene expression profiles of representative cultivated tea accessions.

"There are also some questions need to be clarified

1. Authors only sequenced each tea plant variety for only 5G clean data, which is less than 2 X deep sequencing. it is very not enough for a complex genome with more than 36000 genes annotated, not including many transcript unidentified or annotated.

Under these data size, one may not be able to get SNPs reliable enough, so I doubt about the 930K SNPs are really high-quality?"

Response: We appreciate these technical comments, but believe that they are inconsistent with our data.

First, the comment regarding "less than 2X deep sequencing" is incorrect. This incorrect assessment might have been based on whole genome resequencing data. However, our SNPs were obtained using transcriptome data not genomic data. Transcriptome data only cover transcribed regions, mostly protein-encoding regions, which only account for less than 8% of the tea genome size (total length of the genome is 3,141,536,798 bp, number of annotated genes is 33,932, total length of the genes, including introns, is 250,576,362 bp, accounting for 7.976% of the genome. The percentage would be even smaller if we exclude the introns, which are usually not transcribed). Therefore, our sequencing coverage is about 60X ($15\text{GB}/0.251\text{GB}= 59.8$), not 2X or less.

Second, the comment on "5G clean data" is inaccurate. For each accession we had three biological replicates, thus we obtained around 15 GB of clean data for each accession, not 5GB.

Third, our SNPs are of high quality as explained below. Our transcriptome data covered about 28.6k genes (84.3% of the annotated genes in genome), and on average about 22k genes had a count-per-million-reads (cpm) larger than 1. A total of 17,165,944 SNPs were identified. High quality SNPs were identified based on the following criteria: 1) GATK hard filtering: "QD < 2.0 || DP < 5 || MQ < 40.0 || FS > 60.0 || SOR > 3.0"; 2) only bi-allelic SNPs were retained; 3) only kept SNPs with a missing rate < 20% and MAF > 0.05; 4) no more than 2 SNPs are located within 5 bp from each other. These criteria are in agreement with the criteria used to call high quality SNPs in other crops ^{26,27}. Therefore, we believe that the 925,854 SNPs that we obtained and used in our study truly are of high quality.

"2. Since the population study is set for tea plant evolution, author have not used several other plants in Tea section, and wild tea trees that could help distinguish sub-species, such as CSA and CSS."

Response: We do agree that it is necessary to include some wild tea relatives for the population study, and in fact that is what we did. A wild tea accession (S159, *Camellia taliensis*) was sampled when we collected samples from the major tea growing areas. In addition, the RNA-seq data of other four close relatives of tea (*C. japonica*, *C. azalea*, *C. nitidissima*, and *C. reticulata*) were also obtained from the GEO database and used to call SNPs. These five tea relatives were used as the outgroup to build our phylogenetic tree and helped reliably classify sampled accessions into different clades.

"3. This study confirmed the common observation, CSA- also known large-leaf tea plants, and CSS- also known as small leaf tea plants, and some middle-leaf tea plants as either adaptive to warm or cold environments or hybrids of CSA-CSS. CSA plants had more flavonols, EGCG catechins, and hydrolysable tannins, as compared with CSS. but this study did not give out their specific reasons for that, even certainly, phenylpropanoid genes most likely are more highly expressed in CSA than in CSS, as indicated by previous studies."

Response: With due respect, we disagree with the reviewer's opinion that our results simply "confirmed the common observation...". In fact, our study encompasses several novel findings/observations at the combined metabolomic and population levels that were unprecedented in tea research and reveal metabolite signatures for the five different tea populations: (1) This is the first untargeted metabolomic study of fresh leaf samples involving a large number of diverse representative tea accessions (136); (2) Our untargeted metabolomic analyses have uncovered many compounds not previously described in tea; (3) Our metabolomic analyses discovered many more compounds that exhibited differential accumulation in the leaves of *sinensis*, *assamica* and intermediate accessions than the few phenolic compounds analyzed in the previous measurements; (4) most importantly, our untargeted analyses involving a large number of representative tea accessions have revealed signature metabolites for each of the five phylogenetic groups. The only paper that describes a large-scale comparison of metabolite contents between CSA and CSS tea accessions was Jin et al., in which only the catechin contents of 403 tea accessions collected from various locations in China were measured and compared. The authors reported that the mean total catechin contents in three tea varieties, *sinensis* (n=320), *assamica* (n=21) and *publimba* (n=30), were 152.9±16.2 mg/g, 162.8±22.3 mg/g and 165.1±21.3 mg/g, respectively²⁸ and that the average contents of EGC, C, EC, and ECG in CSA were higher than those in CSS. However, it only focused on catechin compounds. Our metabolomic results agreed with this study on the content differences of ECG, EC and C, but not of EGC. More importantly, our study presented a whole picture of metabolite content differences among different tea populations and many findings were not reported before. For example, we found that many other flavonoid metabolites such as flavonol glycosides, proanthocyanidin dimers and hydrolysable tannins showed large variations between CSA and CSS lineages. Detailed metabolite content differences among different tea populations were presented in our study in terms of differentially accumulated metabolites (DAMs) and signature metabolites. Lastly and most

importantly, we have identified metabolite signatures for the five tea phylogenetic groups (see above).

We agree with the reviewer that it would be important to understand the reasons/mechanisms behind the metabolic differences among the major groups of tea accessions. However, the conclusive study of these mechanisms is clearly beyond the scope of this study, although our work here provides some useful information as discussed below. Regulations of metabolite types and levels are very complicated and may occur at posttranslational, translational, transcriptional, post-transcriptional level or even epigenetic level. Several studies have been examined the expression changes of catechin biosynthesis-related genes in tea plant mutants^{29,30,31}, or plants collected from different seasons³², and the results indicated that the expressions of some catechin biosynthesis-related genes were highly related to the catechin content in tea plants, as one may expect, but this type of study is far from mechanistic understanding that requires integrated omic and genetic studies. Nonetheless, we used the available data to conduct some analyses as follows: (1) we analyzed copy number variations for genes that are known to be involved in the biosynthesis pathway of the three major metabolites in tea, i.e. catechin, caffeine, and theanine. These genes were annotated when the two tea genomes were published^{33,34}. We found no evidence in copy number variation for these genes. (2) we examined whether metabolite biosynthesis genes are located in the selective sweep regions. We did find a few such genes, indicating that some metabolic genes may be subject to strong selection during evolution and domestication of tea. However, the enrichment of these genes is not statistically significant and thus we cannot rule out that they are located in the selective sweep regions by chance. (3) we found on 8,187 SNP loci, CSA accessions possess a different major allele from that of CSS accessions, with some of those SNPs located on metabolite biosynthesis genes (e.g. *LAR* and *TCS*). Some of these SNPs represent non-synonymous mutations that could cause change of protein sequences, which could lead to difference in enzyme activity (as we discussed in the main text, CSA and CSS populations appear to possess different major alleles for the *TCS1* gene, whose enzyme activities have been shown to differ significantly). (4) we identified differentially expressed genes (DEGs) among different tea groups and checked whether any of the known biosynthesis genes of the three major metabolites were differentially expressed. We did find 31, 9, and 5 DEGs that are involved in the catechin, caffeine, and theanine biosynthesis pathway, respectively. For example, seven caffeine biosynthesis genes showed higher levels of expression in CSA accessions, while some catechin biosynthesis genes were highly expressed in different tea groups. However, the expression levels of these genes did not necessarily show direct correlation with the accumulation levels of the related metabolites, indicating that much of changes in the metabolite level may be regulated at the post-transcriptional level. We added a section in the Discussion regarding this point.

To identify potential genes that determine the molecular mechanisms behind the regulation of tea metabolites, an extensive and in-depth genome-wide association analysis of genomic SNPs, transcriptomic and metabolomic data from a large

number of samples, and the construction of gene regulation network, and mGWAS (as suggested by the reviewers) will be necessary. This will be an extremely labor-intensive and expensive undertaking requires multi-year and multi-group efforts. Therefore, this is not within the scope of our study focusing on determining metabolic signatures of representative tea accessions.

"4. Although it is well-known that the accumulation of plant specialized metabolites, including tea plant characteristic metabolites, such as catechins, theanine, caffeine, and various volatiles, mostly are results from plants suffering various abiotic and biotic stresses, and there are highly variable relationships between certain metabolites and their corresponding synthetic genes or regulation factor genes, it is still possible to draw some reliable and solid relations between key gene expression with these metabolites. Authors should step further to draw the relationships, by using more other data, either public or their own experimental data."

Response: We do agree that abiotic and biotic stresses may have significant effects on the accumulation of various metabolites in tea plants and that it is possible to establish some links between the expressions of known synthetic genes and the corresponding metabolite content. In fact, most key structural genes involved in the synthesis of catechins, theanine, and caffeine are already known (as illustrated in the two genome papers). On the other hand, the regulators (especially transcription factors) that are involved in these processes are largely unknown. As discussed above, this is beyond the scope of our current study.

"5. Authors somewhat had mistaken these tea plant genetic variations, tea processing suitability, with local people 's traditional tea processing favorites. Because green teas can be made in all provinces with the tea plant leaves harvested from their local tea plant varieties. so do other types of teas, such as Oolong. The famous types of Oolong teas in Fujian mostly is due to their sophisticated Oolong tea processing techniques, their marketing results, or other non-plant scientific factors. This may not be due to their specific metabolites, or genetic backgrounds. The Oolong teas now are also produced by other provinces by using local tea leaves with similar quality proved that points."

Response: The reviewer is absolutely correct in that genetic background is different from tea processing suitability. For example, 'Tieguanyin' tea cultivars can be processed into both oolong and black tea products. 'Longjing 43' is historically used to produce green tea but it can also be made into oolong tea with good quality. 'Fuding Dabai' has a good processing suitability for making not only white tea, but also other types of tea products. However, through domestication and cultivation or simply by tradition, people have realized that tea cultivars of certain genetic background are most suitable for making certain types of processed tea products. For example, CSA tea plants are most suitable for making pu'er and black tea products, and oolong tea cultivars are most suitable for making oolong tea products. This does not mean that these tea leaves cannot be processed into different types of tea products. Indeed, several publications have already addressed this point,

showing that tea processing suitability has a lot to do with metabolite differences in different cultivars ^{33, 35, 36}. Our classification of tea groups (especially green tea and oolong tea accessions) was based on which type tea product each accession is most likely/suitable to be processed into.

"6. Authors identified some evolutionary selection sweep regions, among more than 800 selection sweep regions, two of them contained caffeine synthetic gene AMPDA, and tri-hydroxylated catechins, these may suggest caffeine and catechins are evolved at least through these sweeps. However, metabolite profiling indicated that among 136 tea varieties, catechins changed significantly, in both quantities or quality. but caffeine is almost similar in content, similar to theanine. Why these controversial results were obtained?"

Response: With due respect, we do not agree that these results are controversial. First, a gene was selected during evolution/domestication does not necessarily mean it was selected for higher expression (i.e. "being selected" is not equal to "higher expression"). It could be selected for higher or lower expression (if mutations occurred in the regulatory regions). Or it could be selected for change of gene function (e.g. non-synonymous mutations that caused change of protein sequence, which in turn caused change of gene function such as enzyme activity).

Second, as we discussed in earlier responses, expression levels of certain biosynthesis genes do not necessarily show direct/strong correlation with the corresponding metabolite levels.

Third, the fact that caffeine and theanine contents did not show as much variations as that of catechin does not mean that caffeine and theanine pathways cannot be selected during tea evolution/domestication. Caffeine and theanine are main compounds that define tea taste and flavor and must be maintained at a level that is liked by tea consumers. Furthermore, for some caffeine- and theanine-genes, there may not be direct correlation between gene expression level and metabolite content level.

Finally, metabolite-related genes could be located in the selective sweep regions due to hitchhiking effect. i.e. they are in the selective sweep regions because neighboring genes/regions but not themselves have been selected.

In addition, compensatory effect may be a factor. Most genes in plant genomes are in a multi-gene family and they may work together to perform the same function. Both genes (*F3'5'H-TEA026294* and *AMPDA-TEA017069*) that we found to be in selective sweep regions belong to multi-gene families and neither of them had the highest expression among genes in the same family. This means that these two genes may not be the main contributor/determination factor of the corresponding metabolite level, but rather serve as a compensatory factor.

"7. While the title of the manuscript is about metabolite signature of tea *Camellia*

sinensis population, the metabolite profiling, identification, and classification were far behind these of transcriptome analysis.

Tea plants contain numerous secondary metabolites with diverse structures and physiochemical properties, it is a big challenge for most labs to properly profiling and precise identification of these tea plant specialized metabolites, as these done in rice and maize, it has to be realized that no enough number of identified specialized metabolites certainly limited the in-depth understanding of tea plants, since these are their most significant features.

Particularly, these differentially accumulated metabolites between CSA and CSS, as well as among CSS subgroups. However, only 57 annotated metabolites among 355 positive ion peaks, or 76 metabolites with identified names among 286 negative ion peaks. As tea plants are featured with their diverse and complex metabolites, more annotation work on these metabolites should help to illustrate more or real signature metabolites, among these tea populations."

Response: We would like to thank the reviewer for this constructive suggestion. Indeed, better annotation of metabolites is of critical importance for the comprehensive study of plant metabolites. Following this suggestion, we performed a thorough re-annotation of the metabolic features discovered in this project. Among the 503 metabolic features that had a relative abundance over 500 in at least one accession under NEG mode, the number of annotated features is 486 (97%) and the number of unannotated features is 17 (3%). Under POS mode, the numbers of total, annotated, and unannotated are 752, 735 (98%), 17 (2%), respectively. We also found that many previously-unannotated features were actually fragment ions that were produced by the MS fragmentation process. Number of fragments is 228 (45%) under NEG mode, and is 556 (74%) under POS mode. We did increase the number of metabolites with clear identity to 230 (46%) under NEG mode and 151 (20%) under POS mode. After removing fragment ions, we found 199 and 129 differentially accumulated metabolites (DAMs) (among which numbers of unannotated DAMs are 11 and 9 respectively) under NEG and POS mode, respectively. Therefore, our annotation of metabolites has been significantly improved and we revised our manuscript accordingly (please see pages 12-17, Table 2, Table 3, Fig. 4, Supplementary Tables S3-S7). For comparison purpose, we reviewed the annotation results from some recent untargeted metabolic profiling of fresh tissues in other plants. Zhu et al. used LC-MS/MS-based metabolic profiling to analyze the metabolite content of fruits from 442 tomato accessions³⁷. A total of 980 metabolic features were detected, among which 362 (37%) features were putatively annotated. Notably, those annotated features included not only the specialized metabolites, but also a lot of primary metabolites such as standard amino acids, nucleic acids, vitamins and their derivatives, which are usually easier to detect and annotate. Liu et al. used LC-MS to conduct untargeted metabolic profiling of 151 tobacco leaf samples from 54 accessions and identified and annotated 158 metabolites³⁸. Similarly, LC-MS/MS analysis of rice leaves from 529 accessions detected 840 metabolites with 277 being annotated³⁹ (including some

primary metabolites). Using LC-ESI-MS/MS to profile wheat grains from 182 accessions, 805 metabolites were detected and 387 of them were annotated, including primary metabolites ⁴⁰. In wheat leaves of 179 doubled haploid lines, Hill et al. obtained 558 metabolic features, of which 197 were putatively annotated ⁴¹. Therefore, compared to these studies, our updated annotation of the metabolites is at least as good, if not better.

"8. The confirmative identification of metabolite features or commonly observed metabolite phenotypes on these tea plant varieties in this study limited the capability of this study to look really insight in the substantial features. The limitation is most likely due to these unreachable or not unveiled metabolites in these tea plant samples or too small size of total tea plant samples."

Response: We are not completely sure what the reviewer is suggesting. We have resolved the metabolite annotation issue (please see our response to point #7) through thorough re-annotation of metabolites. For this study focusing metabolite signatures of different phylogenetic groups, the sample size is not a problem because we have sampled representative accessions in all major tea-growing areas in China and these accessions largely represent the genetic and metabolite diversity of tea plants.

Reviewer #2 (Remarks to the Author):

"The manuscript "Metabolite signatures of diverse *Camellia sinensis* tea populations" by Yu and coworkers provides an interesting and very useful analysis of the metabolites found in a collection of tea germplasm, grown under field conditions at various locations in China. The research is comprehensive and appropriately performed. Some of the interpretations could be improved, but this only requires a rigorous rewrite. I provide a few suggestions, in page order, below. On two general points, the authors should emphasize that the metabolites they analyze are from unprocessed leaves, and are expected to be significantly different from those that would be seen in the processed teas that are actually consumed. I believe that there is quite a bit of literature on this subject, and some of it should be referenced in both the Introduction and the Discussion. The authors also need to tone down their conclusions vis-à-vis the significance of their transcriptome results (see Page 14 comments below) and of their selective sweep results (see Page 9&10 comments below)"

Response: We are very grateful to the reviewer for these constructive suggestions and the extensive specific suggestions to improve the writing. Indeed, analyses of metabolite contents in fresh leaves are very different from those of processed tea products because tea processing can significantly change the contents of many metabolites. We emphasize that fresh leaf samples were used in our study when appropriate and included some references on metabolite content analysis based on processed tea products ^{1, 2, 3, 4} in the Introduction and the Discussion. As suggested, we also tone down our conclusions on the significance of transcriptome results and

selective sweep results. We also cited a few studies ^{42, 43, 44, 45} to show that the metabolite contents can change significantly during tea processing procedures in the Introduction.

Page 2

It is not clear whether the “abundance” refers to metabolites or tea plants. I suggest that the first sentence of the abstract be changed to “....system to study the evolution and diversification of the numerous classes, types and variable contents of specialized metabolites.”

I suggest “which are beneficial for targeted tea breeding and improvement” be deleted. This is a generic sort of statement, and the point was already made on lines 34, 35 and 37 in the Abstract.

Response: Revised as suggested.

Page 3

Line 71: “plant specialized” should be “plant-specialized” because it is a compound adjective.

Line 77: “validations” should be “validation”

Line 78: “at an early stage to answer” is awkward wording, with “cannot yet answer” being a superior replacement

Line 85: “Tea plant” should be “The tea plant”

Response: Revised as suggested.

Page 4

Line 88: “that offers” should be “and offers”

Line 88,89: “anti-cardiovascular and anti-allergic activities” should be “anti-cardiovascular disease and anti-allergy activities”

Line 100: Actually, I believe that theanine is produced in many plants, and even some fungi, and is not unique to tea. It is particularly abundant in tea.

Line 107: It is not proven that China is the place where tea originated, although the area between Assam, Yunnan and Burma seems most likely at this time. It is clear that tea was first officially domesticated in China. By the way, the correct phrasing is “center of origin” not “origin center”

Response: Revised as suggested. Special thanks for pointing out that theanine is also found in fungi.

Page 6

Line 155: “natural variations” should be “natural variation”, while “and maintained and propagated” should be “and are mostly propagated”

Line 174: To what does “wild tea” correspond? Wild tea should mean *Camellia sinensis* that was not planted or otherwise maintained by humans. Are you calling these other *Camellia* species “wild tea”? If so, that is incorrect and should be replaced by “tea relatives” or “close tea relatives”

Line 177: “comprising exclusively of CSA type accessions” should be “containing exclusively CSA accessions”

Response: Revised as suggested. Thanks for the suggestions. S159 is a close tea relative (*C. taliensis*). S9 (Chaoyang) was derived from a wild tea plant (i.e. uncultivated tea plant).

Page 7

Line 182: What does “were adopted from wild tea plants” actually mean? Do the authors mean “adopted” or “adapted”? All domesticated tea originally was adapted from wild tea plants, so it is unclear what is meant by these authors in the case presented here.

Line 196: “using” should be “using the”

Line 199: “from the crossing between” should be “from a cross between”

Response: Revised as suggested. “Adapted” means that they were taken from the wild population and propagated asexually without artificial breeding or cross.

Page 9

Line 243: “sweep” should be “sweeps”

Line 255: “by natural environment” should be “by natural environments”

Lines 262-269: Regarding the candidate selective sweep regions, is there any enrichment for genes that are candidates for known tea quality metabolites? Of course, with 833 regions, you will find some with genes involved in tea leaf metabolites just by chance, but is there any enrichment? Depending on how one sets the criteria, potential selective sweeps can always be “found” but that does not mean they are anything other than chance regions of low polymorphism

Line 270: “indicated” is way too strong, and would be better replaced by “suggested”

Response: Revised as suggested.

To determine if there is any enrichment of tea quality metabolites related genes in selective sweep regions, we tried the following:

First is GO term enrichment analysis. However, there are very limited number (~5000) of genes in the current tea genome database that are annotated with GO terms. There are a few enriched GO terms that appear to be related to tea quality metabolites: ribose phosphate metabolic process, pectin biosynthetic process, proteasome-mediated ubiquitin-dependent protein catabolic process, gene expression, urine nucleotide metabolic process, nicotinamide nucleotide metabolic process, pyrimidine-containing compound metabolic process, carbohydrate catabolic process, glucose 6-phosphate metabolic process. However, it is hard to draw a clear conclusion from these terms. Therefore, we chose not to include these results.

Second, we did a Fisher’s exact test. The total number of annotated genes in the tea genome is 33,932, number of genes in the selective sweep regions is 1,132, number of genes known to be related to tea quality metabolites (genes known to be involved in the pathway of caffeine, theanine, and catechin) is 116, and among these, 3 are

located in the selective sweep regions. A Fisher's exact test gave a p-value of 0.74, which is not statistically significant. Thus, we cannot rule out these the three tea metabolites-related genes were found in the sweep regions was simply by chance.

Page 10

Line 272: as indicated by my comments for lines 262-270, I think that "were the" should be replaced by "may have been"

Lines 281-284: If, as the M&M says, these leaves were fast frozen (as in ref 20), then this needs to be stated somewhere here. This was not a metabolite analysis of processed tea, but rather of tea leaves.

Lines 295-300: What was the statistical significance analysis on this? A multiple testing correction (e.g., Bonferroni) would mean that differences would need to be very high and very consistent for any of these results to be established as significant.

Response: Revised as suggested. We used the Benjamini-Hochberg (BH) method to adjust p-values for multiple comparisons (as described in the M&M section).

Page 11

Line 323: "spectroscopy" should be "spectroscopic"

Response: Revised as suggested.

Page 12

Line 344: "was featured with" should be replaced by "consistently exhibited"

Line 361: "twice of that" should be "twice that" while "accessions" should be "accessions,"

Response: Revised as suggested.

Page 14

Lines 402-418: The authors make much of the differences observed at the transcription level for these genes, but page 11 results indicated that they did not vary much at the level of the metabolites produced. This point is then made, very appropriately, by the authors on lines 418-422. However, just saying this is "rather complex" is too little. I think the Discussion needs a place where it is stated that these results indicate that much of the regulation of metabolite levels occurs post-transcriptionally, and thus it is not clear how often (or if ever!) the transcriptional results may be informative relative to significant differences between the cultivars.

Response: Revised as suggested.

Page 15

Lines 425-426: The first sentence is generic and obvious, so it should be deleted.

The second sentence is not good English and somewhat begs the question of the entire study, so "The rich constitutes of specialized metabolites are essential" should be replaced by "The rich constitution of special metabolites in the growing tea leaf are believed to be essential"

Line 432: "polymorphism" should be replaced by "polymorphic"

Response: Revised as suggested.

Page 16

Lines 460-461: Are somewhat obvious statements, poorly worded and not necessarily true. That is, how long has the divergence of CSS and CSA types been ongoing? If it is just a few thousand years, then that is unlikely to be enough time for significant random drift/divergence. I would suggest that the lines “resulted from long-term divergence and natural or artificial selections” be replaced by “resulting from natural and/or agronomic selection”

Line 462: “are featured with” should be replaced by “have”

Lines 465-467: I expect this is true, but where are the references to support these ideas? If you cannot reference the ideas, then this just be stated as a possibility, not as a certainty.

Response: Revised as suggested. There are two very different estimates on the divergence time between CSA and CSS. One estimate (22,000 years) was based on microsatellite data ⁴⁶ and the other (over 380,000 years) was based on genome-wide SNP markers ³⁴ ³⁴.

Page 17

Line 504: I believe the authors mean “reticulata” not “reticulate” and “from the public database” is not sufficient unless the authors tell us which public database(s) they are referring to.

Response: Revised as suggested. Public database refers to the GEO database and we added it to the text.

Page 18

Lines 523-533: It would be good to provide more information on how heterozygosity was called, and also to have a supplementary table that describes overall heterozygosity for each accession investigated

Line 543: “run for 10” should be “run 10”

Response: Revised as suggested. Details on variants-calling and filtering for high-quality SNPs were added to the Method section. Because heterozygosity values may vary among different SNP sites and a gene may be unevenly transcribed from the two parental copies, we did not describe overall heterozygosity for each accession. Genome sequencing data may be better for calculating heterozygosity for each accession.

Page 19

Line 547: “with 10-kb” should be “with a 10-kb”

Line 552: The authors show some inconsistency in the manuscript in how they report DNA fragment sizes. On this line, they have written “10-kb” and “1kb”. They need to decide on whether they will use the “-” always or never in this manuscript for this type of designation, and then be consistent throughout.

Lines 553-554: This statistical analysis is key, but most readers will not be knowledgeable in this pi value approach, nor am I clear on the degree to which it takes into account multiple testing nor in how it relates to a statistical value most readers

will understand, like a p-value.

Response: Revised as suggested.

XP-CLR uses allele frequency differentiation at linked loci to detect selective sweeps. Genetic diversity (π) is another method to detect selective sweeps during the domestication of crops ^{47, 48}. In order to reduce false positives, we applied both methods and required that a candidate selective sweep region needed to meet two conditions: (1) it had a top 1% XP-CLR score; (2) it had a top 50% π ratio.

Reviewer #3 (Remarks to the Author):

The manuscript entitled “Metabolite signatures of diverse *Camellia sinensis* tea populations” characterized the population structure and phylogenetic relationships among major tea cultivars and association of tea metabolites with populations using transcriptomic and metabolomic analyses. This is a huge work and obtained significant findings, which will contribute to future targeted tea breeding and improvement. The following points are suggested to be improved.

(1) The abstract need be revised. The precise highlighted findings should be indicated.

Response: Revised as suggested.

(2) Based on the results, the authors suggest that PAL and TCS in different groups contain SNP sites, but further experiments are needed to clarify the functional differences of these SNP sites for PAL and TCS in order to rule out the possibility of synonymous SNPs. Whether or not these SNPs affect the differences in enzyme activity or other properties of the two enzymes, thus affecting the differences in the contents of certain metabolites in different groups.

Response: We would like to thank the reviewer for this constructive suggestion. In the revised manuscript, we decide to use *LAR* (TEA027582) and *TCS* (TEA015791) as examples instead because these two genes had highest expression level among the genes in the same family (thus likely play a predominant functional role) and there were previous studies linking their sequence variations to enzyme activity. We did functional annotation of these SNPs and indicated which SNPs are non-synonymous in the figure. We also added a brief discussion on the possible effect of these SNPs on enzyme activity in the context of previous studies.

(3) The authors suggest that the regions of AMPDA and F3'5'H may be selected, but the corresponding downstream metabolites need to be determined in different populations to show that the two genes are indeed selected in different populations.

Response: AMPDA and F3'5'H are upstream enzymes in the caffeine and catechin biosynthesis pathway, respectively. The immediate product of AMPDA is IMP

(inosine monophosphate) and the immediate product of F3'5'H is dihydrotricetin. We were not able to detect either of these two metabolites in our samples, probably because they are intermediate products in the corresponding pathways and thus are processed immediately into other metabolites by downstream enzymes. Therefore, we cannot compare the abundance of these two metabolites in different populations and thus there is no clear evidence for their selection from this aspect. The abundance of the terminal product for these two enzymes (caffeine and catechin respectively) may be regulated by many factors and at multiple layers, thus it is hard to establish a solid link between the selection on these genes to the abundance of the corresponding terminal product.

(4) Can the characterization of metabolites be analyzed in association with the SNP previously measured? Possible correlations during the period of speculation.

Response: We would like to thank the reviewer for this constructive suggestion. In an on-going project, we aim to identify potential regulators for the metabolite contents by gene expression-metabolite association analysis, construction of gene regulatory network, and mGWAS, and the results will be presented elsewhere.

(5) What are the possible down-regulation reasons for TEA016601 (FLS), TEA023333 (CHS) and TEA023790 (F3'H) in S164 sample?

Response: Sample S164 was collected from Jiamu Yeyatang tea plantation in Yunnan province, which has high elevation and low temperature. Our clustering analysis indicated that all samples collected from this location, regardless of their genetic background, had an overall gene expression profiles that were quite different from those of samples taken in other locations (Supplementary Fig. S3). Our heatmap analysis indicated that many genes related to metabolic pathways were down-regulated in samples collected from Jiamu Yeyatang plantation. Therefore, the down-regulation of FLS, CHS, and F3'H was due to environmental factors.

References

1. Lee JE, *et al.* Geographical and climatic dependencies of green tea (*Camellia sinensis*) metabolites: A ¹H NMR-based metabolomics study. *J Agric Food Chem* **58**, 10582-10589 (2010).
2. Fraser K, *et al.* Analysis of metabolic markers of tea origin by UHPLC and high resolution mass spectrometry. *Food Res Int* **53**, 827-835 (2013).
3. Ji H, *et al.* Metabolic phenotyping of various tea (*Camellia sinensis* L.) cultivars and understanding of their intrinsic metabolism. *Food Chem* **233**, 321-330 (2017).
4. Fang S, *et al.* Geographical origin traceability of Keemun black tea based on its non-volatile composition combined with chemometrics. *J Sci Food Agric* **99**, 6937-6943 (2019).
5. Chen S, *et al.* Metabolite profiling of 14 Wuyi Rock tea cultivars using UPLC-QTOF MS and UPLC-QqQ MS combined with chemometrics. *Molecules* **23**, 104 (2018).

6. Li CF, *et al.* Comprehensive dissection of metabolic changes in albino and green tea cultivars. *J Agric Food Chem* **66**, 2040-2048 (2018).
7. Fang W, Cheng H, Duan Y, Jiang X, Li X. Genetic diversity and relationship of clonal tea (*Camellia sinensis*) cultivars in China as revealed by SSR markers. *Plant Syst Evol* **298**, 469-483 (2012).
8. Yao MZ, Ma CL, Qiao TT, Jin JQ, Chen L. Diversity distribution and population structure of tea germplasms in China revealed by EST-SSR markers. *Tree Genet Genomes* **8**, 205-220 (2012).
9. Zhao DW, Yang JB, Yang SX, Kato K, Luo JP. Genetic diversity and domestication origin of tea plant *Camellia taliensis* (Theaceae) as revealed by microsatellite markers. *BMC Plant Biol* **14**, 14 (2014).
10. Tan LQ, *et al.* Fingerprinting 128 Chinese clonal tea cultivars using SSR markers provides new insights into their pedigree relationships. *Tree Genet Genomes* **11**, 90 (2015).
11. Meegahakumbura MK, *et al.* Indications for three independent domestication events for the tea plant (*Camellia sinensis* (L.) O. Kuntze) and new insights into the origin of tea germplasm in China and India revealed by nuclear microsatellites. *PLoS One* **11**, e0155369 (2016).
12. Wambulwa MC, *et al.* Insights into the genetic relationships and breeding patterns of the African tea germplasm based on nSSR Markers and cpDNA sequences. *Front Plant Sci* **7**, 1244 (2016).
13. Liu SR, Liu HW, Wu AL, Hou Y, An YL, Wei CL. Construction of fingerprinting for tea plant (*Camellia sinensis*) accessions using new genomic SSR markers. *Mol Breed* **37**, 93 (2017).
14. Zhang Y, Zhang X, Chen X, Sun W, Li J. Genetic diversity and structure of tea plant in Qinba area in China by three types of molecular markers. *Hereditas* **155**, 22 (2018).
15. Xu YX, Shen SY, Chen W, Chen L. Analysis of genetic diversity and development of a SCAR marker for green tea (*Camellia sinensis*) cultivars in Zhejiang province: The most famous green tea-producing area in China. *Biochem Genet* **57**, 555-570 (2019).
16. Fang WP, Meinhardt LW, Tan HW, Zhou L, Mischke S, Zhang DP. Varietal identification of tea (*Camellia sinensis*) using nanofluidic array of single nucleotide polymorphism (SNP) markers. *Hortic Res* **1**, 14035 (2014).
17. Yao MZ, Chen L, Liang YR. Genetic diversity among tea cultivars from China, Japan and Kenya revealed by ISSR markers and its implication for parental selection in tea breeding programmes. *Plant Breed* **127**, 166-172 (2008).
18. Niu S, *et al.* Genetic diversity, linkage disequilibrium, and population structure analysis of the tea plant (*Camellia sinensis*) from an origin center, Guizhou plateau, using genome-wide SNPs developed by genotyping-by-sequencing. *BMC Plant Biol* **19**, 328 (2019).
19. Yamashita H, *et al.* Analyses of single nucleotide polymorphisms identified by ddRAD-seq reveal genetic structure of tea germplasm and Japanese landraces for tea breeding. *PLoS One* **14**, (2019).
20. Wang RJ, Gao XF, Yang J, Kong XR. Genome-wide association study to identify favorable SNP allelic variations and candidate genes that control the timing

- of spring bud flush of tea (*Camellia sinensis*) using SLAF-seq. *J Agric Food Chem* **67**, 10380-10391 (2019).
21. Jiang CK, Ma JQ, Liu YF, Chen JD, Ni DJ, Chen L. Identification and distribution of a single nucleotide polymorphism responsible for the catechin content in tea plants. *Hortic Res* **7**, 020-0247 (2020).
 22. Xu LY, Wang LY, Wei K, Tan LQ, Su JJ, Cheng H. High-density SNP linkage map construction and QTL mapping for flavonoid-related traits in a tea plant (*Camellia sinensis*) using 2b-RAD sequencing. *BMC Genomics* **19**, 018-5291 (2018).
 23. Xia E, *et al.* The reference genome of tea plant and resequencing of 81 diverse accessions provide insights into genome evolution and adaptation of tea plants. *Mol Plant* **13**, 1013-1026 (2020).
 24. Yao MZ, Ma CL, Qiao TT, Jin JQ, Chen L. Diversity distribution and population structure of tea germplasms in China revealed by EST-SSR markers. *Tree Genet Genomes* **8**, 205-220 (2012).
 25. Zhang W, *et al.* Genome assembly of wild tea tree DASZ reveals pedigree and selection history of tea varieties. *Nat Commun* **11**, 3719 (2020).
 26. Huang X, *et al.* Genome-wide association studies of 14 agronomic traits in rice landraces. *Nat Genet* **42**, 961-967 (2010).
 27. Morris GP, *et al.* Population genomic and genome-wide association studies of agroclimatic traits in sorghum. *Proc Natl Acad Sci U S A* **110**, 453-458 (2013).
 28. Jin JQ, Ma JQ, Ma CL, Yao MZ, Chen L. Determination of catechin content in representative Chinese tea germplasms. *J Agric Food Chem* **62**, 9436-9441 (2014).
 29. Xiong L, *et al.* Dynamic changes in catechin levels and catechin biosynthesis-related gene expression in albino tea plants (*Camellia sinensis* L.). *Plant Physiol Biochem* **71**, 132-143 (2013).
 30. Wang L, *et al.* Complementary transcriptomic and proteomic analyses of a chlorophyll-deficient tea plant cultivar reveal multiple metabolic pathway changes. *J Proteomics* **130**, 160-169 (2016).
 31. Song L, *et al.* Molecular link between leaf coloration and gene expression of flavonoid and carotenoid biosynthesis in *Camellia sinensis* cultivar 'Huangjinya'. *Front Plant Sci* **8**, 803 (2017).
 32. Liu M, *et al.* Relationship between gene expression and the accumulation of catechin during spring and autumn in tea plants (*Camellia sinensis* L.). *Hortic Res* **2**, 15011 (2015).
 33. Xia E-H, *et al.* The tea tree genome provides insights into tea flavor and independent evolution of caffeine biosynthesis. *Mol Plant* **10**, 866-877 (2017).
 34. Wei C, *et al.* Draft genome sequence of *Camellia sinensis* var. *sinensis* provides insights into the evolution of the tea genome and tea quality. *Proc Natl Acad Sci U S A* **115**, E4151-E4158 (2018).
 35. Li P, *et al.* Metabolomic analysis reveals the composition differences in 13 Chinese tea cultivars of different manufacturing suitabilities. *J Sci Food Agric* **98**, 1153-1161 (2018).

36. Zhang G, *et al.* Transcriptome and metabolic profiling unveiled roles of peroxidases in theaflavin production in black tea processing and determination of tea processing suitability. *J Agric Food Chem* **68**, 3528-3538 (2020).
37. Zhu G, *et al.* Rewiring of the fruit metabolome in tomato breeding. *Cell* **172**, 249-261.e212 (2018).
38. Liu P, *et al.* Integrating transcriptome and metabolome reveals molecular networks involved in genetic and environmental variation in tobacco. *DNA Res* **27**, dsaa006 (2020).
39. Chen W, *et al.* Genome-wide association analyses provide genetic and biochemical insights into natural variation in rice metabolism. *Nat Genet* **46**, 714-721 (2014).
40. Chen J, *et al.* Metabolite-based genome-wide association study enables dissection of the flavonoid decoration pathway of wheat kernels. *Plant Biotechnol J* **18**, 1722-1735 (2020).
41. Hill CB, *et al.* Detection of QTL for metabolic and agronomic traits in wheat with adjustments for variation at genetic loci that affect plant phenology. *Plant Sci* **233**, 143-154 (2015).
42. Tan J, *et al.* Study of the dynamic changes in the non-volatile chemical constituents of black tea during fermentation processing by a non-targeted metabolomics approach. *Food Res Int* **79**, 106-113 (2016).
43. Dai W, *et al.* Characterization of white tea metabolome: Comparison against green and black tea by a nontargeted metabolomics approach. *Food Res Int* **96**, 40-45 (2017).
44. Wang Y, *et al.* Novel insight into the role of withering process in characteristic flavor formation of teas using transcriptome analysis and metabolite profiling. *Food Chem* **272**, 313-322 (2019).
45. Chen S, *et al.* Non-targeted metabolomics analysis reveals dynamic changes of volatile and non-volatile metabolites during oolong tea manufacture. *Food Res Int* **128**, 108778 (2020).
46. Meegahakumbura MK, *et al.* Domestication origin and breeding history of the tea plant (*Camellia sinensis*) in China and India based on nuclear microsatellites and cpDNA sequence data. *Front Plant Sci* **8**, 2270 (2017).
47. Duan N, *et al.* Genome re-sequencing reveals the history of apple and supports a two-stage model for fruit enlargement. *Nat Commun* **8**, 249 (2017).
48. Liang Z, *et al.* Whole-genome resequencing of 472 *Vitis* accessions for grapevine diversity and demographic history analyses. *Nat Commun* **10**, 1190 (2019).

REVIEWERS' COMMENTS

Reviewer #1 (Remarks to the Author):

The revised manuscript added some in depth analysis and reorganization of some parts of paper. Generally speaking, I still do not see any significant advance or novel discovery reported in this study, to our understanding of tea plant secondary metabolism and genetic basis for the accumulation of these diverse metabolites.

In last review, I have described my general comments on it.

1. Tea plant secondary metabolisms are complex systems, many progress in recent years have been made, which basically enabled us to understand them for quite a lot. Particularly, after genomes of several varieties have been reported. However, only then, we realized that these tea plant specialized metabolites are very complicated. They vary significantly quantitatively (contents) and quality (molecule species), in their biosynthesis, accumulation, storage, degradation, in tissues of various developmental stages and under different conditions.

2. Like in many other plants, the secondary metabolism in tea plants have also been affected by numerous environmental factors (light, temperature, raining, altitude, soils, insects, pathogens, etc), endogenous signaling and hormones (JA, auxin, ABA. etc), or their combinational influences. There are so many varying factors controlling the secondary metabolism that one almost could not mark a metabolite signature in tea plant variety with certainty and enough robust conclusions. That's why, the locations of tea gardens pretty often are critical for various quality tea plant production. Even the same tea plant variety cultivated in different places will give drastically distinct secondary metabolite profiles.

3. These are simply because of the nature of plant secondary metabolites in plant adaptation to their changing environments: their sensitive changes accompanying tea plant growth, development, senescence, and in the life cycle.

Although there are fingerprints of genetic variations between different varieties, which may be linked to the changes in certain secondary metabolites. However, some reasons, such as great variations in secondary metabolite profiling technology, effects of environmental changes, make the distinguishing the differences among the varieties rather difficult.

Just like a recent Nature publication, about genotype-environment interaction in Arabidopsis:

Exposito-Alonso et al., 2019. Natural selection on the *Arabidopsis thaliana* genome in present and future climates. *Nature* 573:126–129. which showed how climate or environments impact in genetic fingerprints of plants.

Plant adaptation to different environments by changing their metabolism, growth, behaviors, and for generations, their genetic variations.

Such a climate or environmental stresses driven selection of course occurs in tea plants, first of all, and primarily, on so many types of defensive and protective secondary metabolites.

Therefore, I believe that metabolic signature is almost transient, not reproducible in many robust ways. Therefore, the conclusions made in this manuscript is full of challenges. That's my most concerns, as what I pointed out in previous review comments. With our growing understanding of tea plant secondary metabolism in depth and width, these so called metabolite signatures must be addressed or marked in very clearly described environmental conditions when studies were conducted.

The universality of the metabolite signatures that were concluded from the studies could vary with conditions. So I am not surprised that this study indeed showed that most transcriptome and metabolomic data were not well correlated.

minor points:

1) in Figure 1A, the marks in the map of China are not complete.

2) The results in Figure 1B and Figure 2 about genetic populations of tea plant varieties,

Why still authors classified them into 5 groups, which is somewhat ambitious, and standing on solid basis. Usually, most people group them into 3 groups, which is more suitable.

4) Overall, the correlation between metabolomic and transcriptomic data were not well established, so I am not impressed or even confused on what does this manuscript really tell us.

Suggestions:

1). Gene expression (transcriptomes) and certain subgroups of metabolite profiling could be used for e-GWAS analysis. Successful applications of the strategy have been shown in past years on plants,

2). Narrowing down the candidate genes from large scale gene expression-metabolite profiling analysis for certain metabolic pathway genes could be easier to set up a solid case for the genetic basis of secondary metabolism.

(3) Then examine and confirm the metabolic functions of targeted genes in your enlarged datasets from these tea plant varieties.

Reviewer #2 (Remarks to the Author):

The authors fully answered all of my suggestions. I also took the time to read the comments of the other reviewers, and the answers/revisions that the authors made to these two reviewers. My judgement is that all of the reviewers' comments were suitably answered, and that the manuscript is now much improved from a very strong starting point.

Reviewer #3 (Remarks to the Author):

The authors have clearly improved their manuscript. I only have minor comments as shown below. It can be considered for publication after the following explanations.

1. In the manuscript, the authors described that AMPDA and F3'5'H were in the selective sweep regions, but what were the changes of these genes during the selective process?
2. In lines 515-518, what was the possible reason that FLS, CHS and F3'H significantly down-regulated in tea sample S164.

Point-by-point responses to reviewers' comments

Reviewer #1 (Remarks to the Author):

The revised manuscript added some in depth analysis and reorganization of some parts of paper. Generally speaking, I still do not see any significant advance or novel discovery reported in this study, to our understanding of tea plant secondary metabolism and genetic basis for the accumulation of these diverse metabolites.

Response: As we stated in our response to the first review, our findings are novel and significant for the following reasons:

- (1) Our study has revealed major signature metabolites for each of the five tea phylogenetic populations.
- (2) Our comprehensive untargeted metabolomic study of fresh leaves from 136 widely distributed tea accessions is unprecedented, and has identified many metabolites that have not been reported in tea plants before and is clearly distinct from all previous tea metabolite analyses.
- (3) Our genome-wide high-quality SNP analyses of tea population structure and phylogenetic relationship involves the largest and most diverse tea populations and deepest genome-wide sequencing data, generating the most clear-cut phylogenetic groupings of tea populations with highest confidence to date.
- (4) The samples used in our study were collected from tea accessions that were grown in field conditions, and thus their metabolic and transcriptomic profiles represent real-time information about tea leaves before they were processed into commercial tea products and reflect the interaction between genetic background and environment. To the best of our knowledge, our project is the first to use such design in the study of natural tea populations.

In last review, I have described my general comments on it.

1. Tea plant secondary metabolisms are complex systems, many progress in recent years have been made, which basically enabled us to understand them for quite a lot. Particularly, after genomes of several varieties have been reported. However, only then, we realized that these tea plant specialized metabolites are very complicated. They vary significantly quantitatively (contents) and quality (molecule species), in their biosynthesis, accumulation, storage, degradation, in tissues of various developmental stages and under different conditions.

Response: We agree with the reviewer that specialized metabolites in tea plants are very complicated and that they may vary significantly in types and accumulation levels in different genotypes, tissues, and growth conditions. To understand these variations and the underlying mechanisms, we started by untargeted metabolite profiling of samples collected from different accessions that were grown in major growing areas in China.

2. Like in many other plants, the secondary metabolism in tea plants have also been affected by numerous environmental factors (light, temperature, raining, altitude, soils, insects, pathogens, etc), endogenous signaling and hormones (JA, auxin, ABA. etc), or their combinational

influences. There are so many varying factors controlling the secondary metabolism that one almost could not mark a metabolite signature in tea plant variety with certainty and enough robust conclusions. That's why, the locations of tea gardens pretty often are critical for various quality tea plant production. Even the same tea plant variety cultivated in different places will give drastically distinct secondary metabolite profiles.

Response: The types and contents of specialized metabolites in tea may be affected by environmental conditions. However, this does not mean that environmental conditions are the only factor that determines the types and contents of metabolites in a tea plant. For example, if we grow different tea varieties in the same garden (same environmental conditions), we are not expecting that these varieties would contain the same types of metabolites and at the same levels. Genetic factors can play a significant role and that is why we were able to identify metabolite signatures for different tea groups.

3. These are simply because of the nature of plant secondary metabolites in plant adaptation to their changing environments: their sensitive changes accompanying tea plant growth, development, senescence, and in the life cycle.

Although there are fingerprints of genetic variations between different varieties, which may be linked to the changes in certain secondary metabolites. However, some reasons, such as great variations in secondary metabolite profiling technology, effects of environmental changes, make the distinguishing the differences among the varieties rather difficult.

Just like a recent Nature publication, about genotype-environment interaction in Arabidopsis: Exposito-Alonso et al., 2019. Natural selection on the Arabidopsis thaliana genome in present and future climates. Nature 573:126–129. which showed how climate or environments impact in genetic fingerprints of plants.

Plant adaptation to different environments by changing their metabolism, growth, behaviors, and for generations, their genetic variations.

Such a climate or environmental stresses driven selection of course occurs in tea plants, first of all, and primarily, on so many types of defensive and protective secondary metabolites.

Therefore, I believe that metabolic signature is almost transient, not reproducible in many robust ways. Therefore, the conclusions made in this manuscript is full of challenges. That's my most concerns, as what I pointed out in previous review comments. With our growing understanding of tea plant secondary metabolism in depth and width, these so called metabolite signatures must be addressed or marked in very clearly described environmental conditions when studies were conducted.

The universality of the metabolite signatures that were concluded from the studies could vary with conditions. So I am not surprised that this study indeed showed that most transcriptome and metabolomic data were not well correlated.

Response: We agree that specialized metabolites may change with growth conditions and the life cycle of tea plants. However, this does not suggest that metabolite signature is “almost transient, not reproducible”. In the first review, the reviewer wrote “this study confirmed the common observation,, CSA plants had more flavonols, EGCG catechins, and hydrolysable

tannins, as compared with CSS” (comment #3). This “common observation” suggests that metabolite signature is not “almost transient, not reproducible”. As explained in our main text, we believe that these metabolite signatures are caused by genetic factors.

minor points:

1) in Figure 1A, the marks in the map of China are not complete.

Response: We use this figure to show the geographical origins of the tea accessions that we examined in this study. In the original Figure, we used grey background to indicate the major growing areas (provinces) that we sampled the tea plants. As suggested by the reviewer, this may cause confusion that the marks in the map of China are not complete. We revised the Figure by removing the grey background.

2) The results in Figure 1B and Figure 2 about genetic populations of tea plant varieties, Why still authors classified them into 5 groups, which is somewhat ambitious, and standing on solid basis. Usually, most people group them into 3 groups, which is more suitable.

Response: We used 45,162 fourfold-degenerate SNPs to classify our tea accessions into five groups with high confidence. This result agrees with results by Yao et al. (Tree Genetics and Genomes 8, 205-220, 2012), in which SSR markers were used to analyze 450 accessions, and with results by Zhang et al. (Nature Communications 11, 3719, 2020), in which SNPs from transcriptome data were used to classify 221 accessions. We noticed that in Xia et al. (Molecular Plant 13, 1013-1026, 2020), 81 tea accessions were classified into three clades (CSS, CSA, and wild type) based on SNPs from genome resequencing data. However, the result by Xia et al. is likely caused by a small sample size (only 81 accessions were sampled, among which only 58 were cultivated accessions) and the limited genetic diversity that these samples represent.

4) Overall, the correlation between metabolomic and transcriptomic data were not well established, so I am not impressed or even confused on what does this manuscript really tell us.

Response: As we explained in the main text, the lack of correlation between metabolic and transcriptomic data suggests that the regulation of metabolites may not occur only at the transcriptional level, but at different levels (transcriptional, post-transcriptional, translational, post-translational, and epigenetic) and many factors may be involved.

Suggestions:

1). Gene expression (transcriptomes) and certain subgroups of metabolite profiling could be used for e-GWAS analysis. Successful applications of the strategy have been shown in past years on plants,

2). Narrowing down the candidate genes from large scale gene expression-metabolite profiling analysis for certain metabolic pathway genes could be easier to set up a solid case for the genetic basis of secondary metabolism.

(3) Then examine and confirm the metabolic functions of targeted genes in your enlarged datasets from these tea plant varieties.

Response: We appreciate these suggestions and we do agree that e-GWAS analysis, expression-metabolite association analysis, and functional characterization of genes involved in metabolite regulation can help improve our understanding of the underlying mechanisms of metabolite diversity in tea populations. However, these analyses are clearly out of scope of this paper.

Reviewer #2 (Remarks to the Author):

The authors fully answered all of my suggestions. I also took the time to read the comments of the other reviewers, and the answers/revisions that the authors made to these two reviewers. My judgement is that all of the reviewers' comments were suitably answered, and that the manuscript is now much improved from a very strong starting point.

Response: We thank the reviewer for his/her detailed suggestions in the previous review that helped improve our manuscript and for his/her overall positive evaluation of the paper.

Reviewer #3 (Remarks to the Author):

The authors have clearly improved their manuscript. I only have minor comments as shown below. It can be considered for publication after the following explanations.

Response: We thank the reviewer for his/her suggestions that helped improve our manuscript and for his/her recommendation for publication.

1. In the manuscript, the authors described that AMPDA and F3' 5' H were in the selective sweep regions, but what were the changes of these genes during the selective process?

Response: We thank the reviewer for the constructive suggestion. Following this suggestion, we examined the genetic changes in the coding regions of these two genes. We found 18 SNPs in the coding region of the AMPDA gene (TEA017069), among which 5 are nonsynonymous. There are quite a few SNPs in the UTR and intronic regions as well. For gene F3'5'H (TEA026294), 31 SNPs were found in the coding region, including 14 nonsynonymous SNPs. Because we did not find significant enrichment of metabolic pathway related genes in the sweep regions and these two genes may be located in the sweep regions by chance (hitchhiking), we did not perform further analysis on these genes. In the revised manuscript, we toned down our conclusion based on these results.

2. In lines 515-518, what was the possible reason that FLS, CHS and F3'H significantly down-regulated in tea sample S164.

Response: Sample S164 was collected from Jiamu Yeyatang tea plantation in Yunnan province, which has high elevation and low temperature. Our clustering analysis indicated that all samples collected from this location, regardless of their genetic background, had an overall gene expression profile that was quite different from those of samples taken in other locations (Supplementary Fig. S3). Our heatmap analysis indicated that many genes related to metabolic pathways were down-regulated in samples collected from Jiamu Yeyatang plantation. Therefore, the down-regulation of FLS, CHS, and F3'H was due to environmental factors (high

elevation and low temperature). In the revised manuscript, we added this explanation to the main text.